# Measurement and modelling of the dynamics of $NH_3$ surface-atmosphere exchange over the Amazonian rainforest

Robbie Ramsay[1,2,a], Chiara F Di Marco[1], Mathew R Heal[2], Matthias Sörgel[3,b], Paulo Artaxo[4], Meinrat O. Andreae[3,5], and Eiko Nemitz[1]

[1]UK Centre for Ecology and Hydrology (UKCEH), Bush Estate, Penicuik, EH26 0QB, UK
[2]School of Chemistry, University of Edinburgh, Joseph Black Building, David Brewster Road, Edinburgh EH9 3FJ, UK
[3]Biogeochemistry Department, Max Planck Institute for Chemistry, 55128 Mainz, Germany
[4]Instituto de Física, Universidade de São Paulo, São Paulo, Brazil
[5]Scripps Institution of Oceanography, University of California San Diego, La Jolla, CA, USA
[a]now at: NERC Field Spectroscopy Facility, James Hutton Road, Edinburgh, EH9 3FE, UK
[b]now at: Atmospheric Chemistry Department, Max Planck Institute for Chemistry, Mainz, Germany

**Correspondence:** Eiko Nemitz (en@ceh.ac.uk)

**Abstract.** Local and regional modelling of $NH_3$ surface exchange is required to quantify nitrogen deposition to, and emissions from, the biosphere. However, measurements and model parameterisations for many remote ecosystems—such as tropical rainforest—remain sparse. Using one month of hourly measurements of $NH_3$ fluxes and meteorological parameters over a remote Amazon rainforest site (Amazon Tall Tower Observatory, ATTO), six model parameterisations based on a bi-directional, single-layer, canopy compensation point resistance model were developed to simulate observations of $NH_3$ surface exchange. Canopy resistance was linked to either relative humidity at the canopy level ($RH_{z_0'}$), vapour pressure deficit, or a parameter value based on leaf wetness measurements. The ratio of apoplastic $NH_4^+$ to $H^+$ concentration, $\Gamma_s$, during this campaign was inferred to be $38.5 \pm 15.8$. The parameterisation that reproduced the observed net exchange of $NH_3$ most accurately was the model that used a cuticular resistance ($R_w$) parameterisation based on leaf wetness measurements and a value of $\Gamma_s = 50$ (Pearson correlation $r = 0.71$). Conversely, the model that performed the worst at replicating measured $NH_3$ fluxes used an $R_w$ value modelled using $RH_{z_0'}$ and the inferred value of $\Gamma_s = 38.5$ ($r = 0.45$). The results indicate that a single layer, canopy compensation point model is appropriate for simulating $NH_3$ fluxes from tropical rainforest during the Amazonian dry season, and confirmed that a direct measurement of (a non-binary) leaf wetness parameter improves the ability to estimate $R_w$. Current inferential methods for determining $\Gamma_s$ were noted as having difficulties in the humid conditions present at a rainforest site.

## 1 Introduction

The global cycling of nitrogen is of critical importance to Earth's biogeochemistry. One of the major contributors to the global atmospheric reactive nitrogen ($N_r$) budget is ammonia ($NH_3$), which is primarily generated from anthropogenic sources (Galloway et al., 2003). The emission of $NH_3$, and the subsequent deposition of $NH_3$ or other forms of reactive nitrogen, have impacts on terrestrial and marine ecosystems (Erisman et al., 2013). In particular, forests can be impacted through changes to N input in several ways. Fowler et al. (2013) detail how increased deposition of N can lead to increased vegetation growth rates in

forests, leading to potentially greater carbon sequestration rates. This potential positive impact, however, is offset by the effect of N saturation on forests as detailed by Nadelhoffer (2008). Here, the combined impact of disturbance to forest soil microbial systems involved in the nitrification-denitrification cycle (Fowler et al., 2009) and damage to vegetation (Krupa, 2003) leads to a sharp decrease in net primary productivity. Even at deposition rates well below the saturation values, atmospheric $N_r$ deposition can lead to changes in plant species composition, with implication not only on biodiversity but also ecosystem services (Fowler et al., 2013). It is therefore important that exchange models be developed for all major biomes to accurately simulate $NH_3$ deposition rates to forests to predict potential environmental consequences.

Ammonia is emitted in small quantities from (semi-)natural sources such as wild fires and the excreta of wild animals, and also from plants as a result of non-zero $NH_4^+$ concentrations within the leaf apoplast, and from decomposing leaf-litter; both plant sources vary with the N status of the plants. Of further importance for nitrogen modelling is therefore the determination of the extent of potential emissions from forest ecosystems and the role such $NH_3$ emission might play in the $N_r$ cycle within natural forests. Forests were once considered to be perfect sinks for ammonia (Duyzer et al., 1992), until bi-directional surface exchange of $NH_3$—i.e., deposition to and emission from—were recorded in many studies of $NH_3$ fluxes from forests (Langford and Fehsenfeld, 1992; Neirynck and Ceulemans, 2008; Wyers and Erisman, 1998). Predominantly, this has been observed in forests situated close to sources of agriculturally derived $N_r$ pollution, although Hansen et al. (2015) also observed bi-directional fluxes over a more remote forest site.

The modelling of regional and local surface exchange of $NH_3$ is based on parameterisations of the exchange, which remain unverified for many biomes of global importance due to the difficulty and cost of making measurements of $NH_3$ fluxes. Datasets of $NH_3$ flux measurements have mainly been limited to temperate agricultural and semi-natural ecosystems. Consequently, very little is known about the role of $NH_3$ in the N cycling in remote ecosystems such as the tropics and their disturbance through anthropogenic activity. Although Flechard et al. (2015) identified the need for $NH_3$ surface exchange measurements over tropical ecosystems, and over rainforests in particular, such measurements have been limited so far. Here we present recent data from the Amazon Tall Tower Observatory site, situated in remote tropical rainforest, where $NH_3$ fluxes were measured for one month during the dry season of 2017 as part of a suite of species (Ramsay et al., 2020). This provides the data necessary to develop site-specific parameterisations of $NH_3$ surface exchange, with the potential for upscaling to the regional level. The companion paper summarises the measured fluxes, including their statistics and average diurnal cycle and discusses the uncertainties associated with the measurement.

As extensive reviews of $NH_3$ surface exchange models are available (Flechard et al., 2015; Massad et al., 2010), only a brief overview is provided here. Models of bi-directional $NH_3$ surface exchange consider the control of fluxes to be analogous to electrical resistances (Baldocchi, 1988; Monteith and Unsworth, 2013). Whether emission occurs from the atmosphere to the canopy or vice versa is dependent upon the relative magnitude of ambient and canopy concentrations, with resistances acting in series or in parallel impeding the exchange. In the simplest model of bi-directional surface exchange, all exchange is approximated to occur via the leaf stomata situated at a single notional mean height (big-leaf approach) and is restricted by two atmospheric resistances in series (the aerodynamic resistance and quasi laminar boundary layer resistance), in series with a third (stomatal) resistance (stomatal compensation point model) (Sutton et al., 1993).

Increasingly complex models include further pathways of exchange (Kruit et al., 2010), with the most important for the current study being the canopy compensation point model, initially proposed by Sutton et al. (1995), which incorporates two parallel pathways of exchange at the canopy level (Figure 1). In the first pathway, a stomatal compensation point ($\chi_s$) is introduced, which represents the concentration of $NH_3$ in the leaf stomata in (temperature dependent) equilibrium with the $NH_4^+$ and pH of the apoplastic fluid. This stomatal compensation point controls the exchange of $NH_3$ to and from the canopy to the leaf stomata, together with the associated stomatal resistance ($R_s$). In the parallel pathway, a unidirectional deposition flux is modelled from canopy to the leaf cuticle, with a separate cuticular resistance ($R_w$) controlling deposition. In a modified version of this model (the cuticular capacitance model), the leaf cuticle is considered to be both a sink and source for $NH_3$ (Sutton et al., 1998). Here, the ability of water films on the leaf cuticle surface to act as a storage of previously deposited $NH_3$ is introduced as an analogue of an electrical capacitor, with emission fluxes of $NH_3$ from the cuticle possible with the evaporation of "charged" water films. Further models include ones which simulate the potential for soil and leaf litter below canopy to act as emission sources of $NH_3$ (Nemitz et al., 2000; Sutton et al., 2009). Using the static canopy compensation point

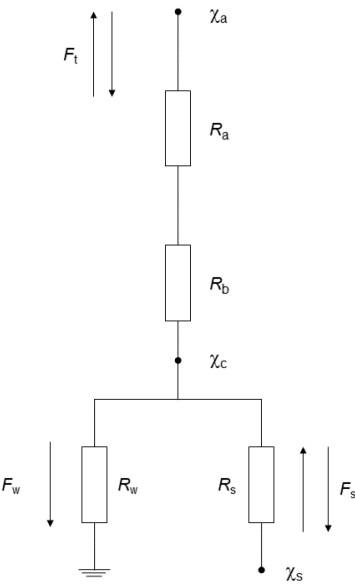

**Figure 1.** Schematic of the canopy compensation point model of Sutton et al. (1995). $F_t$, $F_s$, and $F_w$, are, respectively, the total, stomatal, and cuticular fluxes of $NH_3$; $R_a$, $R_b$, $R_w$, and $R_s$, are, respectively, the aerodynamic, quasi-laminar boundary layer, cuticular and stomatal resistances; and $\chi_a$, $\chi_c$, and $\chi_s$ are, respectively, the atmospheric concentration of $NH_3$, the canopy compensation point, and the stomatal compensation point.

model of $NH_3$ surface exchange, in combination with new $NH_3$ flux and meteorological data measured at a remote, tropical rainforest site, this study aims to present a series of local model formulations for $\chi_s$ and $R_w$ which adequately simulate the bi-directional fluxes of $NH_3$ observed by Ramsay et al. (2020), with focus on the most suitable control metric for $R_w$.

Statistical comparison between models is conducted, with the aim to determine which parameterisation—and hence, which of the factors controlling the formulation of model parameters—is best able to simulate observed fluxes. Discussion includes how meteorological conditions may have influenced model performance, and how subsequent studies of $NH_3$ fluxes over tropical rainforest may be conducted to help improve model performance. We discuss other model frameworks, such as the dynamic CCP model, that could be used to simulate $NH_3$ bi-directional surface exchange in Section 4.1, with a focus on the simplicity and performance of the SCCP model as justification for not pursuing more complex models further.

## 2 Methodology

### 2.1 Field site description

Measurements were conducted on an 80 m walk-up tower located at the Amazon Tall Tower Observatory site ($2° 08.637'$S, $58° 59.992'$W). The ATTO site lies on a level plateau 120 m above sea level, and is situated within a region of dense, undisturbed *terra firme* rainforest, with a mean canopy height of 37.5 m (Chor et al., 2017). The nearest large urban centre, Manaus, Brazil, is located 150 km to the south west. A full description of the ATTO site, its permanent instrumentation and the floristic composition of the surrounding rainforest is given in Andreae et al. (2015).

The rainforest extends homogenously for many hundreds of kilometres to the north and east, but gives way to shrub-forest ("*campina*") 5.5 km to the south, where the plateau descends to meet the Uatumã River. The flux fetch requirement for these gradient measurements with a geometric mean height of 50.2 m as determined from the approximation given by Monteith and Unsworth (2013), is 5.2 km. Therefore, $NH_3$ fluxes can be considered representative of a homogeneous rainforest. During convective daytime conditions, the flux footprint is much shorter, typically $< 2$ km.

Measurements were made between 6 October and 5 November 2017, during the region's dry season. Lasting typically from August to November, the dry season is characterised by warmer, drier conditions in comparison to the wet season which lasts from February to May. Air masses that arrive at the site during the dry season typically travel over some urban and agricultural areas located to the south and south east of the site, which can give rise to periods of elevated black carbon ($BC$) and carbon monoxide (CO) concentrations (Ramsay et al., 2020; Saturno et al., 2018).

### 2.2 Measurements of ammonia and meteorological parameters

#### 2.2.1 Ammonia

Ammonia was measured using the Gradient of Aerosols and Gases Online Registration system (GRAEGOR), a semi-autonomous, continuous wet-chemistry instrument (ECN, The Netherlands) (Thomas et al., 2009). The GRAEGOR provides online analysis of a suite of inorganic trace gases ($NH_3$, HCl, HONO, $HNO_3$ and $SO_2$) and their associated water-soluble aerosol counterparts ($NH_4^+$, $Cl^-$, $NO_2^-$, $NO_3^-$ and $SO_4^{2-}$) at two heights at hourly resolution. The instrument consists of two sample boxes, which were set at two heights ($z_1 = 42$ m and $z_2 = 60$ m) on the 80 m walk-up tower, with a detector box which is connected to each sample box located in an air conditioned container at ground level for online analysis of samples.

Each sample box contains a wet annular rotating denuder (WRD) connected in series to a steam jet aerosol collector (SJAC). A short section of high-density polyethylene (HDPE) tubing connects an inlet cone covered with an HDPE insect mesh to the sample boxes of the WRDs, ensuring that losses of $NH_3$ are minimised. The walls of each WRD are coated in a constantly replenishing sorption solution of 18.2 MΩ double deionised (DDI) water, with 0.6 mL of $H_2O_2$ (9.8 M) added per 10 L of sorption solution to eliminate potential biological contamination of the WRDs. Air is drawn simultaneously through both WRDs at a rate of $16.7 \, L \, min^{-1}$, kept constant by a critical orifice downstream of the WRD. Unlike $NH_4^+$ aerosol, gaseous $NH_3$ diffuses through the laminar air flow onto the sorption solution coating the walls of the WRD, and the solution is subsequently transported to the detector box at ground level for analysis.

The detector box contains a flow injection analysis unit (FIA) based on a selective ion membrane to analyse the concentration of $NH_3$ within the WRD samples. WRD samples are fed to the FIA unit, where NaOH (0.1 M) is first added to the sample to form gaseous $NH_3$. The gaseous $NH_3$ then passes through a semi-permeable polytetrafluoroethylene (PTFE) membrane to enter a counterflow of DDI water to re-form $NH_4^+$. The temperature-corrected conductivity of $NH_4^+$ is then measured in the conductivity cell of the FIA unit, from which the atmospheric concentration of $NH_3$ at the height from which the sample was drawn can be determined. Through a valve control system within the detector box, the WRD sample from each height is analysed for $NH_3$ by FIA once per hour, resulting in an hourly-resolved concentration gradient of $NH_3$. The FIA unit is calibrated autonomously using three liquid $NH_4^+$ solutions (0, 50, and 500 ppb $NH_4^+$ concentration), with the first calibration conducted 24 h after the GRAEGOR begins measurements, and every 72 h afterwards. Fresh standards were prepared prior to each calibration. A total of 10 autonomous calibrations were conducted during this campaign.

### 2.2.2 Meteorology

Wind speed ($u$), wind direction ($wd$), friction velocity ($u_*$) and sensible heat flux were measured by a Gill Windmaster mounted at 46 m on the 80 m walk-up tower. Relative humidity and air temperature were measured at 22 m, 36 m and 55 m using a series of Campbell HygroVUE™5 Temperature and Relatively Humidity Sensors. Net radiation and photosynthetically active radiation were measured at 75 m by, respectively, a net radiometer (Kipp and Zonnen NR-LITE2) and a quantum sensor (Kipp and Zonnen PAR LITE). Hourly rainfall was measured using a HS Hyquist TB4-L.

### 2.3 Modified Aerodynamic Gradient Method

In the constant flux layer over homogeneous surfaces, the flux of a chemical tracer $\chi$ can be determined using the aerodynamic method (AGM) if the vertical concentration gradient of $\chi$ and its diffusion coefficient are known (Foken, 2008). A modified form of the AGM, based on the vertical concentration difference ($\Delta_c$) between measurements of $NH_3$ at 42 m and 60 m, a series of stability parameters determined from meteorological measurements, and $u_*$ as measured at 46 m by eddy-covariance (Flechard, 1998), was used to determine the flux of $NH_3$ as:

$$F_x = -u_* \kappa \frac{\Delta_c}{\ln\left(\frac{z_2 - d}{z_1 - d}\right) - \Psi_H\left(\frac{z_2 - d}{L}\right) + \Psi_H\left(\frac{z_1 - d}{L}\right)} \tag{1}$$

where $\kappa = 0.41$ is the von Kármán constant and $d$ is the zero-plane displacement height, determined as $0.9h_c = 33.4$ m. The integrated form of the heat stability correction term, $\Psi_H$, is included to account for deviations from the log-linear wind profile, while the term $(z\text{-}d)/L$ is a dimensionless measure of atmospheric stability, where $L$ is the Obukhov length.

The aerodynamic gradient method strictly holds for measurements made within the inertial sublayer. Corrections must be applied to fluxes calculated using the AGM if measurements are made close to the canopy, within the roughness sublayer, as was the case in this study. Fluxes were corrected using a correction factor, $\gamma_F$, whose magnitude was determined from the stability conditions present at the time of measurement (Chor et al., 2017). The validity of this correction was confirmed via the flux measurements of $HNO_3$ and $HCl$ by Ramsay et al. (2020).

## 2.4 Determination of concentrations and meteorological parameters at the aerodynamic mean canopy height

The aerodynamic resistance $R_a$ and the quasi-laminar boundary layer $R_b$ can be used to determine the temperature and $NH_3$ concentration at the aerodynamic mean canopy height, $z_0'$, if their respective values at a reference height are known (Nemitz et al., 2009):

$$T_{z_0'} = T(z - d)\frac{H}{\rho c_p}\left(R_a(z - d) + R_b\right) \tag{2}$$

$$\chi_{z_0'} = \chi(z - d) + F_\chi\left(R_a(z - d) + R_b\right) \tag{3}$$

The relative humidity at $z_0'$ can be determined if the saturation pressure at $z_0'$ ($\varepsilon_{\mathrm{sat}}(T_{z_0'})$) and the water vapour pressure at $z_0'$ ($\varepsilon_{z_0'}$) are known. $\varepsilon_{z_0'}$ can be calculated as:

$$\varepsilon_{z_0'} = \varepsilon(z - d) + F_{H_2O}\left(R_a(z - d) + R_b\right) \tag{4}$$

where $F_{H_2O}$ is the measured water vapour flux, as taken at the 80 m tower. $RH_{z_0'}$ is then calculated as:

$$RH_{z_0'} = \frac{\varepsilon_{z_0'}}{\varepsilon_{\mathrm{sat}}\left(T_{z_0'}\right)} \times 100 \tag{5}$$

From measurements of $T_{z_0'}$ and $RH_{z_0'}$, the vapour pressure deficit (VPD) in kPa was determined.

## 2.5 Canopy Resistance Method

The basic resistance model that describes deposition to a non-perfectly absorbing surface approximates the ability of the surface to regulate $NH_3$ deposition through a canopy resistance, $R_c$, which can be calculated from the difference between (a) the total resistance towards deposition (i.e., the inverse of the deposition velocity ($V_d$) of $NH_3$ at a reference height) and (b) the sum of the atmospheric aerodynamic resistance, $R_a$, and the quasi-laminar boundary layer resistance, $R_b$, Fowler and Unsworth (1979); Wesely et al. (1985):

$$R_c = \frac{1}{V_d(z - d)} - R_a(z - d) - R_b \tag{6}$$

$R_a$ and $R_b$ can be determined from Eq. (7) and (8), respectively (Garland, 1977; Ramsay et al., 2020):

$$R_a(z-d) = \frac{u(z-d)}{u_*^2} - \frac{\Psi_H(\zeta) - \Psi_M(\zeta)}{\kappa u_*} \tag{7}$$

$$R_b = (Bu_*)^{-1} \tag{8}$$

where $B$ is the sublayer Stanton number (Foken, 2008). However, this canopy resistance approach as outlined in Eq.(6) can only successfully be applied if there is no bi-directional exchange. Since both emission and deposition of $NH_3$ was observed in this study, a bi-directional exchange model was required to simulate surface atmosphere exchange of $NH_3$. The simplest bi-directional exchange model for $NH_3$ is the static canopy compensation point (SCCP) model (Figure 1) in which the exchange between the canopy and the atmosphere is controlled by a conceptual canopy compensation point ($\chi_c$).

In this SCCP model, the total surface-atmosphere exchange of $NH_3$ ($F_t$) is the sum of two constituent fluxes, the unidirectional deposition of $NH_3$ to the cuticle surface ($F_w$), and a bidirectional flux of $NH_3$ through the leaf stomata ($F_s$) (Sutton et al., 1995):

$$F_t = F_w + F_s \tag{9}$$

where

$$F_w = \frac{-\chi_c}{R_w} \tag{10}$$

and

$$F_s = \frac{(\chi_s - \chi_c)}{R_s} \tag{11}$$

For the stomatal exchange flux, the difference between the notional mean concentration at canopy height (the canopy compensation point concentration ($\chi_c$)), and the stomatal compensation point concentration ($\chi_s$) provides the numerator on the right-hand-side term in Eq. (11). When $\chi_s$ exceeds $\chi_c$, emission occurs. $\chi_s$ is proportional to the ratio of dissolved $NH_4^+$ to $H^+$ in the leaf apoplast (the apoplastic ratio), which represents a dimensionless emission potential ($\Gamma$), via a temperature function that describes the combined Henry solubility and dissociation equilibrium (Nemitz et al., 2004):

$$\chi_s = \frac{161,500}{T} \exp\left(-\frac{10380}{T}\right) \Gamma_s \tag{12}$$

Here, $T$ is the temperature of the canopy in K.

The stomatal resistance, $R_s$, is primarily dependent on global radiation ($S_t$), with additional potential influences from factors such as temperature, vapour pressure deficit, and leaf and root water potentials. Here the generalised function for bulk stomatal resistance as per Wesely (1989) with the parameters recommended for tropical vegetation is used to calculate the stomatal resistance for $NH_3$ ($R_s(NH_3)$):

$$R_s(NH_3) = R_i \left\{ 1 + \left(200(S_t + 0.1)^{-1}\right)^2 \right\} \left\{ 400 \left[ T_{z_0'}\left(400 - T_{z_0'}\right)\right]^{-1} \right\} \tag{13}$$

where $R_i$ represents the minimum bulk resistance stomatal resistance for water vapour (for deciduous forest during summer: $R_i = 70\,\mathrm{s\,m^{-1}}$); $S_t$ is the global radiation in $\mathrm{W\,m^{-2}}$; and $T_{z_0'}$ is the temperature in $^\circ$C at the mean canopy height.

The appropriateness of this parameterisation and the choice of parameter $R_i$ was evaluated against the water vapour fluxes that were measured during fairly dry conditions when stomatal evapotranspiration is expected to be the dominant source.

The bulk stomatal resistance ($R_{sb}$) for water exchange can be calculated from the measured water vapour flux ($F_{\mathrm{H_2O}}$) as

(Nemitz et al., 2009):

$$R_{sb} = \frac{\varepsilon_{\mathrm{sat}}\left(T_{z_0'}\right) - \varepsilon_{z_0'}}{F_{\mathrm{H_2O}}} \tag{14}$$

To avoid periods during which sources other than evapotranspiration contribute to the water flux we applied a stringent filter criterion to exclude periods during or within two hours of rainfall or with $RH > 80\%$. This left 55 30-minute values for the assessment. Measurement-derived values of $R_{sb}$ for $\mathrm{H_2O}$ were converted to the equivalent resistance for $\mathrm{NH_3}$, accounting for

the differences of the molecular diffusivities of the two gases (e.g. Hanstein et al. (1999)), and the comparison was carried out on their reciprocal values (stomatal conductances, $G_s$ and $G_{sb}$), because it is the uncertainty in the stomatal conductances that propagates directly into the predicted flux. A linear regression analysis revealed a very high $R^2$ value of 0.97 and a slope of 0.95 (using an intercept of 0), suggesting that the modelled resistances were slightly larger, but well within the range of the measurement uncertainty of $R_{sb}$. Therefore, the parameterisation based on Wesely (1989) is appropriate for this site.

This parallel cuticular pathway in the SCCP model treats the flux to the leaf cuticle ($F_w$) to be unidirectional to a perfectly absorbing sink, given by the ratio of the canopy compensation point ($\chi_c$) and the cuticular resistance ($R_w$). $R_w$ has been described successfully by a number of empirically derived parameterisations in various studies as outlined by Massad et al. (2010), with most using either relative humidity or water vapour pressure deficit as proxies for the ability of $\mathrm{NH_3}$ to absorb to the leaf surface. The term $R_w$ is discussed further in Section 3.4.

The canopy compensation point ($\chi_c$) is the conceptual mean concentration of $\mathrm{NH_3}$ inside the canopy, at which the stomatal, cuticular and above-canopy fluxes balance each other. It is therefore dependent upon the ambient concentration of $\mathrm{NH_3}$ ($\chi_a$) and various physical and chemical parameters, both on the surface of the leaf and the surrounding atmosphere, as described by the resistances (stomatal, cuticular, aerodynamic and quasi-laminar boundary layer) and the stomatal compensation point previously described. In this study, $\chi_c$ was calculated as:

$$\chi_c = \frac{\chi_s \times R_w \times (R_a + R_b) + \chi_a \times R_w \times R_s}{R_w \times R_s + R_a + R_b \times R_w + R_a + R_b \times R_s} \tag{15}$$

Prompted by the observation of morning emissions of $\mathrm{NH_3}$ which could not be explained by stomatal exchange alone, this model was further extended by Sutton et al. (1998) to account for bi-directional exchange with leaf surfaces, by allowing $\mathrm{NH_3}$ to be absorbed and desorbed to/from leaf water layers. The extended model calculated the $\mathrm{NH_3}$ holding capacity by estimating the leaf water amount in relation to $RH$. The ammonia holding capacity was implemented into the resistance framework by

220 considering it analogous to an electric capacitor. The charge of this "capacitor" depended dynamically on previously deposited $\mathrm{NH_3}$, and tended to be released as dew dried out in the morning. Similarly, Nemitz et al. (2001) extended the model by a second model layer to describe additional exchange with the ground level or soil.

## 2.6 Leaf Wetness Measurements

Leaf surface wetness was measured using a sensor array as described in Sun et al. (2016), which was based on the design by Burkhardt and Eiden (1994). Six sensors arranged in pairs of two, each consisting of gold-plated electrodes arranged as a clip, were each attached to a leaf situated 27 m above ground level and within the canopy surrounding the 80 m walk-up tower. Each clip provided a measurement in mV that was related to the electrical conductivity between the two electrodes. Data were recorded using a Raspberry Pi Model 2 B (Raspberry Pi Foundation, Cambridge) at a temporal resolution of one minute. The sensor array was checked daily to ensure good contact between the clips and the leaf. Leaf wetness was measured from 6 October to 5 November 2017. Unlike conventional (binary) wetness grid sensors, this approach provides some gradation between fully dry and fully wet canopies.

Raw values from each sensor pair were converted to a leaf wetness parameter value, ranging from 0 to 1, according to the methodology outlined by Klemm et al. (2002). During periods of significant precipitation ($\geq$0.1 mm rainfall per hour), the leaf is considered fully wet and the raw signal from the sensor is at a maximum value. During prolonged dry periods the leaf is considered to be dry, and the lowest recorded conductivity of the sensor pair during these periods is designated as a "zero signal". The net signal from each sensor pair is determined by subtracting the corresponding zero signal from the raw signal for each period of data considered. The cumulative time period of precipitation is then determined from rainfall measurements. For this study, precipitation occurred during 15% of the total campaign time. Consequently, the signal percentile for each sensor that represents periods of precipitation was 85% in this study. Finally, the zero corrected net signals are divided by the value of signal percentile to give a leaf wetness parameter (LWP) whose values range from 0 (dry) to 1 (wet).

## 3 Results

### 3.1 Temperature, relative humidity, VPD and LWP at canopy

The time series of calculations of $T_{z_0'}$, $RH_{z_0'}$ and $VPD_{z_0'}$, together with measurements of the leaf wetness parameter and measured and modelled fluxes of $NH_3$ (see Section 3.2 below), are shown in Figure 2. The measurements can broadly be split into four distinct periods of warmer, drier conditions and cooler, wetter conditions. Period One, from 6 to 18 October, is typified by an average leaf wetness at the canopy of 0.7, with an average RH of 82%, suggesting the prevalence of humid, wet conditions. Period Two extends from 19 to 25 October, where leaf wetness at the canopy decreases, while VPD increases, which is paired with a drop in average $RH$. Conditions resume the same pattern as Period One during Period Three, which lasts between 26 October and 1 November, but gives way to drier, warmer conditions (Period Four) from 2 November until the end of the campaign. A distinct lag exists between the relative humidity at the canopy level and the leaf wetness measurements, particularly during the drier conditions from 19 to 25 October. RH minima, which occur on average between 11:00 and 13:00, are not reflected in leaf wetness measurements until several hours later. Minima leaf wetness measurements during this period are recorded between 13:00 and 16:00.

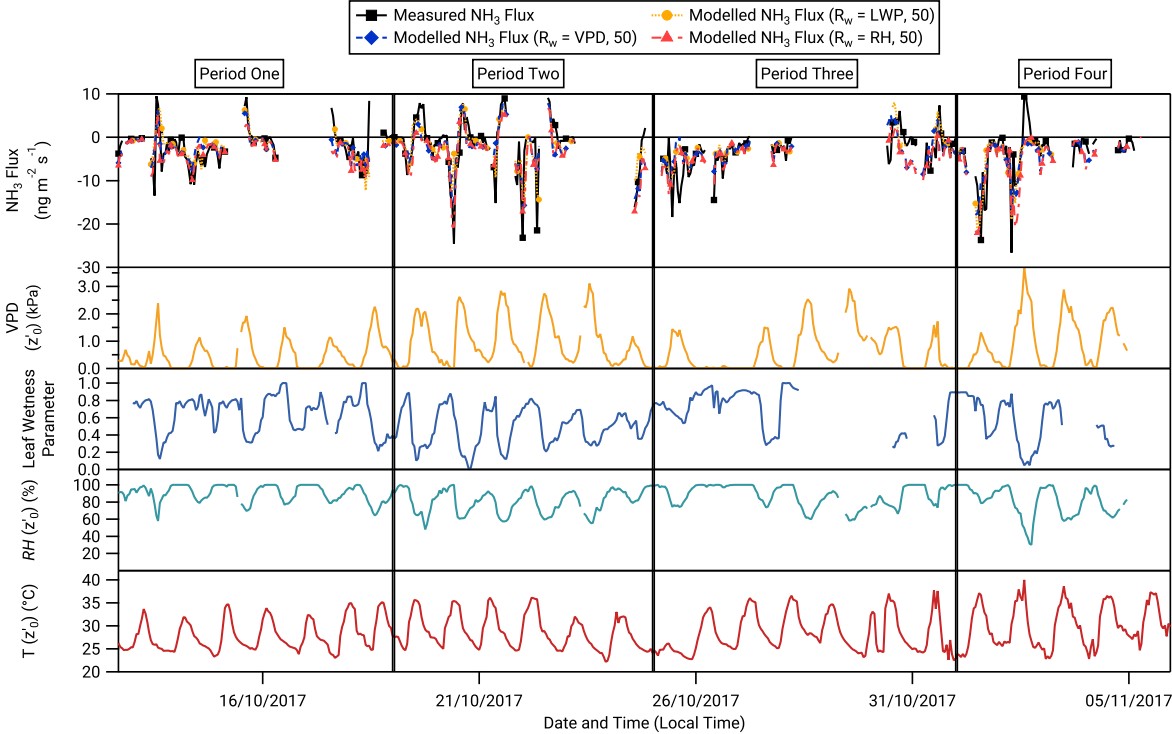

**Figure 2.** Time series of (from top to bottom) measured and modelled $NH_3$ fluxes, vapour pressure deficit at $z'_0$, leaf wetness parameter, relative humidity at $z'_0$, and air temperature at $z'_0$ throughout the period of $NH_3$ flux measurements.

### 3.2 Overview of $NH_3$ measurements

Figure 2 shows the time-series of the measured fluxes , together with model results (see below). Both (positive) emission and (negative) deposition fluxes were recorded during the campaign, ranging from +9.5 $ng\,m^{-2}\,s^{-1}$ to ¬30.2 $ng\,m^{-2}\,s^{-1}$. Figure 3 presents calculated $NH_3$ fluxes as scatterplots for the duration of the campaign against paired canopy values of temperature and relative humidity as well as the leaf wetness parameter. Shaded contour areas—from green to red for low to high density of measurements—are added to the scatterplots to highlight temperature, relative humidity and leaf wetness conditions where

measurements of $NH_3$ fluxes were particularly concentrated. A statistical summary of linear regression models for calculated $NH_3$ flux and the respective parameter plotted is given for each subplot in Figure 3.

While emission and deposition occur across the full range of temperatures recorded during the campaign, with a weak correlation ($R^2 = 0.02$; $p = 0.02$) in the linear regression model of $NH_3$ flux against temperature, suggesting that emissions were more likely to be observed during warmer conditions. Relative humidity appears to be a somewhat stronger driver of

265 $NH_3$ surface exchange behaviour ($R^2 = 0.08$; $p = 3.98 \times 10^{-4}$) than temperature. The slope and density contours suggest that emissions are more likely as relative humidity decreases. The strongest predictor of the three meteorological parameters investigated is the leaf wetness parameter ($R^2 = 0.19$; $p = 2.72 \times 10^{-5}$). Emissions predominantly occur during periods when

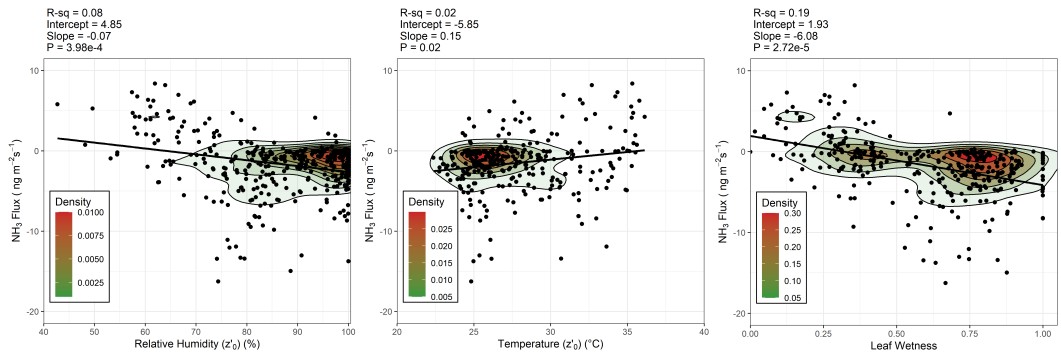

**Figure 3.** Scatterplots with line of best fit and data density shadings for $NH_3$ flux against (from left to right) relative humidity at $z'_0$, temperature at $z'_0$, and leaf wetness parameter.

leaf wetness parameter values fall below 0.5, with deposition occurring predominately during periods when the leaf surface is wet ($>0.6$) or completely saturated (1).

## 3.3 Determination of stomatal compensation points and emission potentials

One of the elements of modelling of $NH_3$ flux through the static canopy compensation point model is the stomatal flux ($F_s$), which, from Eq. (11), depends on the canopy concentration of $NH_3$ ($\chi_c$) and the stomatal compensation point ($\chi_s$). The value of $\chi_s$ is determined by the leaf surface temperature (in this study, taken as $T_{z'_0}$) and the apoplastic ratio ($\Gamma_s$). If $\Gamma_s$ is known, which varies with vegetation type (Hoffmann et al., 1992; Mattsson et al., 2009), environmental stressors such as drought (Sharp and Davies, 2009) and nitrogen nutrition (Massad et al., 2008), then $\chi_s$ and subsequently $F_s$ can be modelled. The fact that emissions at this site regularly occurred during midday (when $T_{z'_0}$ is at its maximum) and were related to drier, warmer conditions (Figure 3) is consistent with the emission flux originating from the stomata.

$\Gamma_s$ can be inferred from measurements during conditions where the $NH_3$ surface exchange is judged to be dominated by stomatal exchange, with a negligible contribution from desorption of $NH_3$ from the leaf surface.

Under conditions where $R_w$ is very large compared with $R_s$, the ambient $NH_3$ concentration ($\chi_a$) at which a zero net flux occurs (i.e., when the difference between $\chi_c$ and $\chi_s$ is 0) is implicitly equal to $\chi_c$ and $\chi_s$. Therefore, if $NH_3$ surface exchange is driven by stomatal exchange, $\chi_s$ may be determined from the values of $\chi_a$ at which the flux changes from deposition to emission, or vice versa (Nemitz et al., 2004). Figure 4 presents the ambient $NH_3$ concentrations measured during the campaign at which such flux sign changes occurs as a function of $T_{z'_0}$, for conditions under which $R_w$ is expected to be fairly large ($RH < 60\%$).

Eq.(12) can therefore be rearranged to give an expression for $\Gamma_s$ that is dependent upon $T_{z'_0}$ and $\chi_s$, where $\chi_s$ can be substituted with a value of $\chi_a$ at which a sign change in the flux of $NH_3$ occurs:

$$\Gamma_s = \frac{1}{\left[\frac{161500}{T_{z'_0}+273} \exp\left(-\frac{10380}{T_{z'_0}+273}\right)\left(\frac{1}{\chi_s}\right)\right]} \tag{16}$$

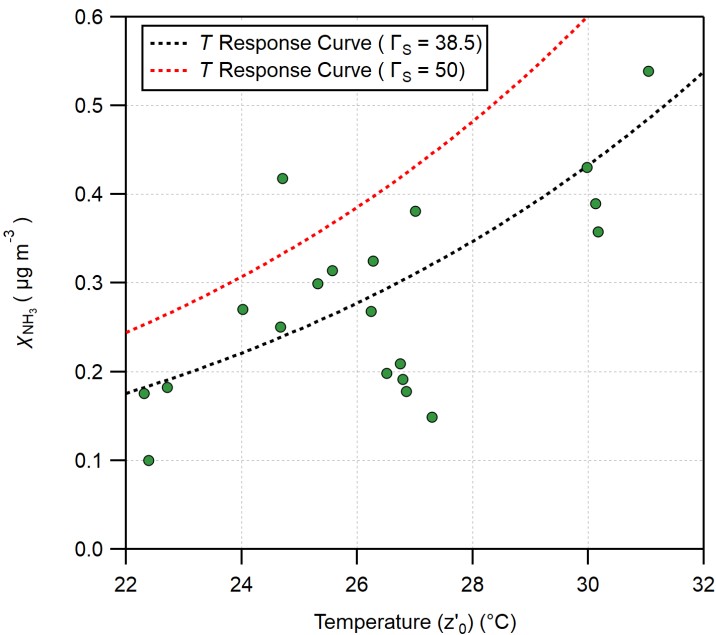

**Figure 4.** Estimating the stomatal compensation point from the ambient $NH_3$ concentrations at which the flux changed signs as a function of the temperature at z′0. The black dotted line shows the temperature response curve calculated using an apoplastic ratio of 38.5, while the red dotted line shows the temperature response curve calculated using a ratio of 50.

Using the values of $\chi_a$ measured in this campaign that are inferred to be equal to $\chi s$, the apoplastic ratio applicable to the period of measurement was determined as $38.5 \pm 15.8$. Also shown in Figure 4 is the temperature response curve of $\chi_s$ consistent with this value of $\Gamma$. Consequently, modelled values of $\chi_s$ based on a value of $\Gamma_s = 38.5$ were determined for the campaign period, and subsequently used to determine values of $\chi_c$ and total modelled flux. As this value resulted in an under-prediction of the peak emissions, an alternative, enhanced value of $\Gamma_s = 50$ was also explored to develop further parameterisations for comparison. By using an enhanced value for $\Gamma_s$, all emissions from the leaf surface are implicitly considered to originate from leaf stomata, rather than cuticular desorption or other potential sources of $NH_3$ emissions, such as soil or leaf litter.

### 3.4 Determination of $R_w$ parameterisations based on three alternative proxies for leaf water volume

Considering the observed drivers for $NH_3$ surface exchange discussed in Section 3.2, three different parameterisations for the cuticular resistance $R_w$ were developed for this study, based upon three alternative proxies of the $NH_3$ holding capacity of the leaf water layers: $RH_{z'_0}$, $VPD_{z'_0}$, and leaf wetness. Subsequently, each parameterisation of $R_w$ was used to develop three distinct values for $F_w$, the unidirectional flux component of the cuticular-resistance-based single-layer model, each describing the surface atmosphere exchange of $NH_3$ at the ATTO site.

The first parameterisation was based on measurements of $RH_{z_0'}$ using the following equation (Sutton et al., 1993):

$$R_w = \alpha + \exp\left(\frac{100 - RH}{\beta_1}\right) \tag{17}$$

Here, $\alpha$ determines the minimum cuticular resistance (which is $\alpha = 1\ \mathrm{s\,m^{-1}}$), while $\beta_1$ is a constant scaling coefficient controlling the increase of $R_w$ with decreasing relative humidity. The coefficients $\alpha$ and $\beta_1$ were fitted by least-squares optimisation between total modelled and observed values of $NH_3$ flux taken during the campaign to arrive at values of $\alpha = 2\ \mathrm{s\,m^{-1}}$ and $\beta_1 = 9$, which were used for modelling $R_w$ based on $RH_{z_0'}$ for the entirety of the campaign. The second parameterisation was based on measurements of the vapour pressure deficit, using a formulation for $R_w$ based upon that employed by Flechard et al. (1999):

$$R_w = \alpha + \beta_2 \left(\exp^{\gamma_1(\mathrm{VPD})}\right) \tag{18}$$

As with the parameterisation of $R_w$ in Eq. (17), the coefficient $\alpha$ is the minimum value for $R_w$ at zero VPD, set at $2\ \mathrm{s\,m^{-1}}$. $\beta_2$ and $\gamma_1$ are constant scaling coefficients which, respectively, control the scaling of the exponential term and the scaling of the vapour pressure deficit response. Through least-squares optimisation, a value of 5 was chosen for $\beta_2$ and 1.7 for $\gamma_1$, for the determination of $R_w$ based on Eq. (18) for the entirety of the campaign.

Finally, a novel parameterisation for $R_w$ based upon measurements of leaf wetness was developed for this campaign based on least-squares optimisation:

$$R_w = \alpha + \beta_3 \left[\exp^{\gamma_2(1-LWP)} - 1\right] \tag{19}$$

As with the parameterisations of $R_w$ described in Eq. (17) and (18), $\alpha$ is the minimum value of $R_w$ at maximum leaf wetness, set at $2\ \mathrm{s\,m^{-1}}$, $\beta_3$ is a scaling coefficient, similar to that of the parameterisation in Eq. (18), set at a value of 5, and $\gamma_2$ is a scaling coefficient controlling the increase in $R_w$ with the decrease in leaf wetness, set in this study to 4.8. With this parameterisation, $R_w$ approaches $\alpha$ for a fully wet canopy and is capped at $R_w = 605\ \mathrm{s\,m^{-1}}$ for a fully dry canopy.

### 3.5 Comparison of modelled with observed $NH_3$ fluxes

Six discrete model runs of $NH_3$ surface exchange were investigated, via Eqs. (9), (10) and (11), using the two values of the apoplastic ratio ($\Gamma_s = 38.5$ and $\Gamma_s = 50$) combined with one of the three parameterisations for $R_w$ (relative humidity $RH$, vapour pressure deficit VPD, and leaf wetness parameter LWP) described above. These models are: a. ($R_w = $ LWP, $\Gamma_s = 38.5$), b. ($R_w = $ LWP, $\Gamma_s = 50$), c. ($R_w = RH$, $\Gamma_s = 38.5$), d. ($R_w = RH$, $\Gamma_s = 50$), e. ($R_w = $ VPD, $\Gamma_s = 38.5$), and f. ($R_w = $ VPD, $\Gamma_s = 50$).

Table 1 presents a summary of the model results, with average mean modelled fluxes for the overall campaign, together with day (06:00–17:00) and night-time (18:00–05:00) average mean values for the four separate periods of the campaign discussed in Section 3.1. Also presented are average mean values for calculated fluxes based on $NH_3$ measurements during the campaign, and the percentage of modelled values which agree in flux direction with observed values. Modelled mean values highlighted in bold signify where the model value deviates by more than 25% from the corresponding observed mean. The models which

**Table 1.** Summary of model results, with comparison to measured hourly $NH_3$ fluxes (in $ng\,m^{-2}\,s^{-1}$). Presented are the overall mean average flux for measured and modelled $NH_3$ fluxes (with associated errors), the correctness of modelled flux direction in comparison to the measured values, and mean average values for modelled and measured $NH_3$ fluxes during day (06:00 – 17:00) and night (18:00 – 05:00) for the four periods of the measurement campaign. Values in bold signify average model values which differs by $\pm$ 25% from corresponding measured average value.

| | Overall | Correctness of direction (%) | Period One Day | Period One Night | Period Two Day | Period Two Night | Period Three Day | Period Three Night | Period Four Day | Period Four Night |
|---|---|---|---|---|---|---|---|---|---|---|
| Measured | -2.83 ± 0.94 | - | -2.80 | -1.47 | -3.36 | -2.14 | -4.74 | -1.97 | -6.61 | -2.02 |
| Model a. $R_w$, LWP, 38.5 | -3.03 ± 0.48 | 87.2 | -3.19 | -1.86 | -3.33 | **-2.81** | -4.55 | **-2.58** | -5.99 | -2.90 |
| Model b. $R_w$, LWP, 50 | -2.69 ± 0.49 | 90.6 | -2.36 | -1.86 | -3.25 | **-2.80** | -3.93 | -2.37 | -5.94 | -2.89 |
| Model c. $R_w$, RH, 38.5 | **-3.97 ± 0.51** | 82.4 | **-4.76** | **-2.45** | **-5.05** | **-3.74** | **-6.35** | **-3.52** | **-4.28** | **-2.76** |
| Model d. $R_w$, RH, 50 | **-3.73 ± 0.56** | 86.8 | **-4.14** | **-2.45** | **-4.15** | **-3.73** | **-5.89** | **-3.52** | **-4.22** | **-2.75** |
| Model e. $R_w$, VPD, 38.5 | -3.48 ± 0.62 | 84.8 | -3.45 | **-2.28** | -3.55 | **-3.58** | -4.62 | **-3.32** | -5.83 | -3.35 |
| Model f. $R_w$, VPD, 50 | -3.16 ± 0.61 | 89.1 | -2.64 | **-2.28** | **-2.50** | **-3.58** | -4.01 | **-3.32** | -5.78 | **-3.34** |

differ least and most from the observed values are models b and c, respectively. Models which use the calculated apoplastic ratio of 38.5 differ more from the observed values than those which use an apoplastic ratio of 50 (near 1 standard deviation from calculated). Modelled daytime values tend to differ less from their corresponding observed value in comparison to night-time values. Modelled values during Period One have the least divergence from the observed overall (day and night), while the greatest divergence in modelled values occurs during Period Four, particularly at night. For model b, 91% of values agree with the observed direction of fluxes, the best performing model using this parameter. Conversely, model c values agree with only 83% of the observed direction of fluxes.

Figure 2 shows the full time series of modelled $NH_3$ fluxes from models b, d, and f alongside the measured flux. In general, periods of emission and deposition are modelled well by all three models, with two exceptions: the emission period from 12:00 to 16:00 on 30 October, where modelled fluxes suggest an earlier emission from 11:00 which lasts for fewer hours; and from 13:00 to 15:00 on 2 November, where no model predicts an emission, in contrast to the measurements. The magnitude of modelled fluxes generally agrees well with the observed flux, although model d, which uses an $R_w$ parameterisation based on $RH$, tends to estimate smaller emissions in comparison to model f ($R_w$ = VPD). Model b ($R_w$ = LWP) comes closest to replicating the magnitudes of the measured emissions, although as with the other two models it underestimates the magnitude of the measured deposition.

### 3.6 Error in observed and simulated fluxes

The error in the observed fluxes of $NH_3$ during this campaign are presented in Ramsay et al. (2020). Using a Gaussian error propagation approach, a median percentage error of 33% was calculated for the observed $NH_3$ fluxes.

The error in the simulated fluxes can also be determined using an error propagation method. Eq.(9) highlights that the total, simulated flux ($F_t$) is the sum of the cuticular flux ($F_w$) and the stomatal flux ($F_s$). The total uncertainty in $F_t$, $\sigma_{F_t}$, can therefore be determined by:

$$\sigma_{F_t} = \sqrt{\sigma_{F_w}{}^2 + \sigma_{F_s}{}^2} \tag{20}$$

where $\sigma_{F_w}$ and $\sigma_{F_s}$ are the associated errors in, respectively, $F_w$ and $F_s$. In turn, $\sigma_{F_w}$ and $\sigma_{F_s}$ can be calculated using a Gaussian error propagation based on Eq.(10) and Eq.(11) respectively, which rely on the errors in $R_a$, $R_b$, $R_s$ and $R_w$ measurements.

While the error in $F_s$ will remain the same for all simulated values of $F_t$, the error in $F_w$ will vary on the choice of $R_w$ parameterisation. Thus, the error in $R_w$ is the primary variable which governs the differences in the error in $F_t$ between the 360   simulated values.

Using this framework, the error values were calculated for each $R_w$ and $\Gamma_s$ parametrisation for the simulated total flux. In the overall calculated total flux in Table 1, the associated error, in $\mathrm{ng\,m^{-2}\,s^{-1}}$, is also shown.

## 4   Discussion

### 4.1   Temporal dynamics

The observed bidirectional surface exchange of $NH_3$ from a remote tropical rainforest site was modelled using a series of canopy compensation point, cuticular resistance based models using a variety of different $R_w$ parameterisations and apoplastic ratios.

As highlighted in the Introduction, measurements of $NH_3$ surface exchange over natural ecosystems such as forests remain sparse. This is particularly true for measurements over remote environments such as tropical vegetation. To our knowledge, 370   there have not been any direct flux measurements over tropical vegetation to date; although Trebs et al. (2004) and Adon et al. (2010; 2013) inferred fluxes from single point concentration measurements. Exchange of $NH_3$ has been measured previously at temperate forest sites and reported to be bi-directional: for example, Langford and Fehsenfeld (1992) noted daytime emission from a remote forest near Boulder, Colorado; Neirynck and Ceulemans (2008) observed bidirectional $NH_3$ exchange (with median diel emissions recorded between 12:00 and 16:00) over a Scots Pine (*Pinus sylvestris*) forest in Flanders, Belgium; 375   while Wyers and Erisman (1998) noted daytime emissions from Douglas Fir (*Pseudotsuga menziesii*) forest at Speuld in the Netherlands. When using models to determine the drivers of surface exchange above forest sites, these studies often stress the importance of cuticular desorption as a further process that dominated in the morning when emission could not have originated from stomatal compensation points. Indeed, in the case of Neirynck and Ceulemans (2008), the static canopy compensation point model (SCCP) was unable to simulate their observed emissions.

At ATTO, there is no indication that the single layer SCCP model was unable to reproduce the temporal dynamics of the measured $NH_3$ surface exchange. Indeed, an exploratory application of the dynamic CCP model did not result in an improvement in model performance and therefore the modelling work here focused on the static model as a simpler approach

able to reproduce the measurements. The absence of morning desorption peaks at the ATTO forest is likely due to the small night-time adsorption of $NH_3$ into leaf water layers associated with the low night-time $NH_3$ concentrations at this site. Dry deposition of aerosol ammonium nitrate ($NH_4NO_3$) is another source of volatile $NH_4^+$ on leaf surfaces, and concentrations of this compound are again typically very small in Amazonia (Wu et al., 2019). In addition, given the high $RH$, the water layers may not dry out as rapidly and completely as at other sites. The measured median $NH_3$ atmospheric concentration at the canopy height during the measurement period was $0.23\ \mu eq\ m^{-3}$, with an estimated annual total reactive N dry deposition input of $1.74\ kg\ N\ ha^{-1}\ yr^{-1}$ (Ramsay et al., 2020). This is far lower than reported by Neirynck and Ceulemans (2008) and by Wyers and Erisman (1998), both of whose sites were subject to high levels of agricultural pollution. As noted by Massad et al. (2010) and Zhang et al. (2010), higher atmospheric inputs of N to forest systems lead to an increase in the stomatal emission potential. Conversely, with lower atmospheric $NH_3$ concentrations, the potential for forests to act as a source of $NH_3$ is increased, as the likelihood of the canopy compensation point exceeding the ambient concentration increases. The low N status of the tropical vegetation may also favour transfer of N absorbed to the cuticle into the leaf, e.g., via liquid films that extend from the cuticle into the stomata (Burkhardt et al., 2012).

In general, the fluxes measured at ATTO could also be reproduced without including a further soil layer, potentially with one exception (see below). Such a layer is needed where night-time emissions are observed that are clearly not under stomatal control (Nemitz et al., 2000; Hansen et al., 2017).

The consistently warmer noon-time conditions at the leaf canopy during measurements at the ATTO site would also favour stomatal exchange driven emissions of $NH_3$. An increase in leaf temperature, with VPD controlled for, has been shown to lead to greater gas exchange through increased stomatal openings (Urban et al., 2017), while alterations to the Henry and dissociation equilibria would lead to a change in the stomatal compensation point favouring increased stomatal emissions. Similarly, the unstable conditions at noon above the canopy over tropical rainforest leads to reductions in $R_a$, which would increase any emissions occurring at the time that were driven by stomatal exchange (Flechard et al., 2015). In the study of forest $NH_3$ emissions that is most similar in ambient $NH_3$ concentrations, canopy compensation points and apoplast ratios to this study, Hansen et al. (2017) comes to a similar conclusion on the observed daytime emissions from a remote, temperate forest in Indiana, USA.

Despite the low N inputs and apoplastic $NH_4^+/H^+$ ratio, significant emission periods were observed above the ATTO site. One driver is clearly the high daytime leaf temperature. The average flux amounted to a small deposition of $\neg 2.8\ ng\ m^{-2}\ s^{-1}$ suggesting that, on average, the site receives more N as $NH_3$ than it loses. Possible sources include small-scale farming and biomass burning.

## 4.2 Apoplast Ratio

The apoplastic ratio of $NH_4^+/H^+$ ($\Gamma_s$) inferred from the measurements in this study was $38.5 \pm 15.8$; the models investigated used either a value of 38.5 or 50 (close to one standard deviation from inferred value). Both values are significantly lower than the majority of $\Gamma_s$ values obtained for other forest sites. Wang et al. (2011) gives a value of $\Gamma_s = 400$ for green temperate forest canopies, which is also used by Hansen et al. (2017). Massad et al. (2010) reviews a range of $\Gamma_s$ values derived from

measurements of $NH_3$ surface exchange over forest, which range from $\Gamma_s = 27$ (as measured directly through bioassay for unfertilised Spruce forest) to $\Gamma = 5604$ as determined by Wyers and Erisman (1998) for a highly polluted *P. menziesii* forest. The study by Neirynck and Ceulemans (2008) used a value of 3300 in spring and 1375 in summer for a *P. sylvestris* forest.

The disparity in the emission potentials (the apoplastic ratios) between other forest sites and the tropical rainforest site at ATTO is again linked to nitrogen input. With larger N inputs where nitrogen is deposited in excess, the stomatal concentration is increased (Schjoerring et al., 1998). Consequently, at polluted areas such as the forest sites studied by Neirynck and Ceulemans (2008), apoplastic ratios are increased, while at sites with lower N input, such as semi-natural vegetation with low ambient $NH_3$ concentrations, values of $\Gamma_s$ can be as low as 5–10 (Hanstein et al., 1999). The species of vegetation is also critical

(Mattsson et al., 2009). Plants which are reliant on mixed nitrogen sources (ammonium, nitrate, and organic N), and which are more reliant on root rather than shoot assimilation of nitrogen, exhibit lower apoplast ratios than nitrate reliant, shoot assimilating species (Hoffmann et al., 1992). The value of 38.5 which was inferred from measurements lies in the range of $\Gamma_s$ values exhibited by semi-natural vegetation with low N inputs, and in the lower range of overall forest values quoted by Massad et al. (2010).

## 4.3   Model Performance with respect to $R_w$ parameterisation

An assessment of the performance of the individual parameterisations against calculated $NH_3$ fluxes is included in Figure 5, which displays the results of simple linear regression models for the simulated values of each $NH_3$ flux model against observed $NH_3$ fluxes. With regards to the Pearson correlation coefficient ($r$), the rank of models from most strongly correlated with observed $NH_3$ fluxes to least correlated is model b, model a, model f, model e, model d, model c (model descriptions are

435 found in Table 1). The $R_w$ parameterisation was a stronger determinant of model-measurement correlation than the choice of $\Gamma$. Correlation with measurements is highest for the models using an $R_w$ based upon LWP, followed by those which use VPD and finally $RH$. Within each grouping, models using $\Gamma = 50$ provide simulated values that have a better correlated fit with observed values than $\Gamma = 38.5$. Overall model performance is in many ways more sensitive to $R_w$ than $\Gamma$ as is apparent from the Taylor diagram (Figure 6) which summarises in one diagram the three complementary model-measurement performance statistics of

440 (i) correlation coefficient ($r$), (ii) centred root mean squared error (RMSE), and (iii) within-model and within-measurement standard deviations (SD) (Taylor, 2001). The statistical metrics visualised in Figure 6 are summarised in Table 2.

    The standard deviation in the observed flux dataset is 3.65 ng m$^{-2}$ s$^{-1}$. The model that comes closest to replicating this same variability is model b (2.65 ng m$^{-2}$ s$^{-1}$), with model e (2.24 ng m$^{-2}$ s$^{-1}$) replicating observed values least well, although the overall range between model standard deviation is broadly similar. It should be borne in mind, however, that the standard

deviation of the measured flux includes measurement uncertainty in addition to real variability, as do the modelled values, which are based on measured parameters. With regards to $r$, model b simulated values produce the highest correlation with the observed values at $r = 0.71$, in comparison to model c, which performs the worst at $r = 0.45$. Finally, the model with the lowest root mean square error from the observed is model b at 2.79 ng m$^{-2}$ s$^{-1}$, with the highest error found in model c, at 3.31 ng m$^{-2}$ s$^{-1}$. Therefore, from the ability of the parameterisations to reproduce the measured average fluxes as presented in

Figure 5, 6, and (Table 1), it can be concluded that parameterisation b, in which the value $R_w$ is parametrised using leaf wetness

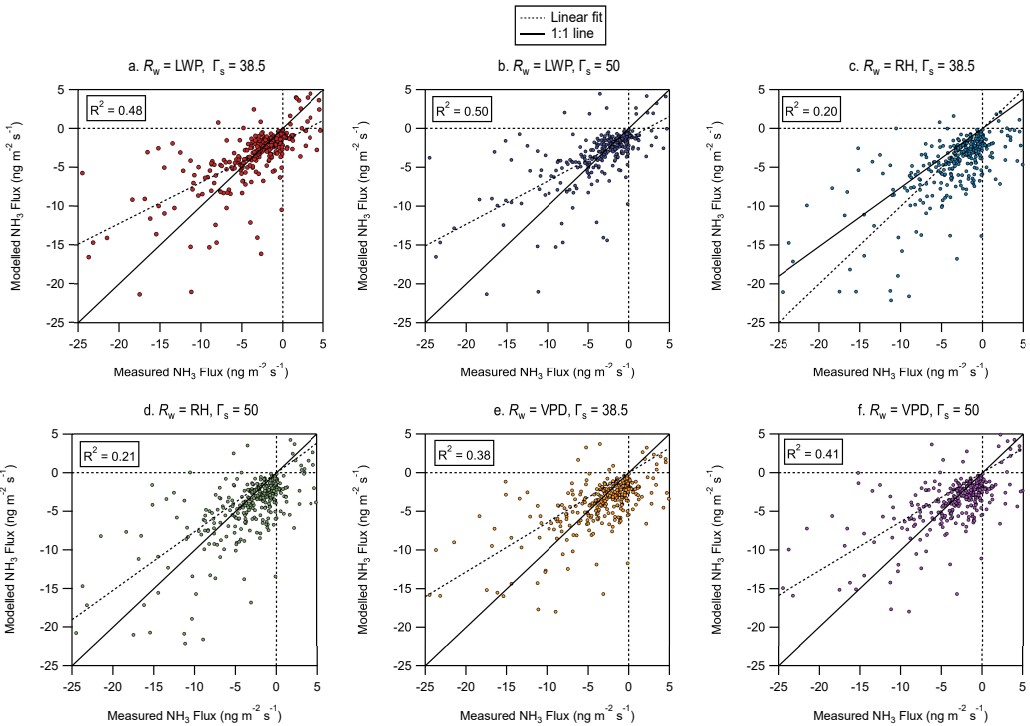

**Figure 5.** Linear regressions between measured $NH_3$ fluxes and six models (a–f) of $NH_3$ fluxes differing in the approach used to derive a value for the cuticular resistance $R_w$ and in the value used for the apoplastic ratio $\Gamma$, as noted above each panel.

parameter values and where the apoplastic ratio is set to 50, is the best performing model in simulating $NH_3$ surface-atmosphere exchange at the ATTO site, while model c is the worst performing.

**Table 2.** Summary of model statistical performance (correlation coefficient R, centred root mean square error, and standard deviation) as presented in Figure 6.

| | $R$ | RMSE ($\mathrm{ng\,m^{-2}\,s^{-1}}$) | Standard Deviation within model ($\mathrm{ng\,m^{-2}\,s^{-1}}$) |
|---|---|---|---|
| Model a. $R_w$, LWP, $\Gamma$ = 38.5 | 0.69 | 2.81 | 2.49 |
| Model b. $R_w$, LWP, $\Gamma$ = 50 | 0.71 | 2.79 | 2.65 |
| Model c. $R_w$, RH, $\Gamma$ = 38.5 | 0.45 | 3.31 | 2.60 |
| Model d. $R_w$, RH, $\Gamma$ = 50 | 0.46 | 3.27 | 2.62 |
| Model e. $R_w$, VPD, $\Gamma$ = 38.5 | 0.62 | 2.92 | 2.24 |
| Model f. $R_w$, VPD, $\Gamma$ = 50 | 0.64 | 2.90 | 2.49 |

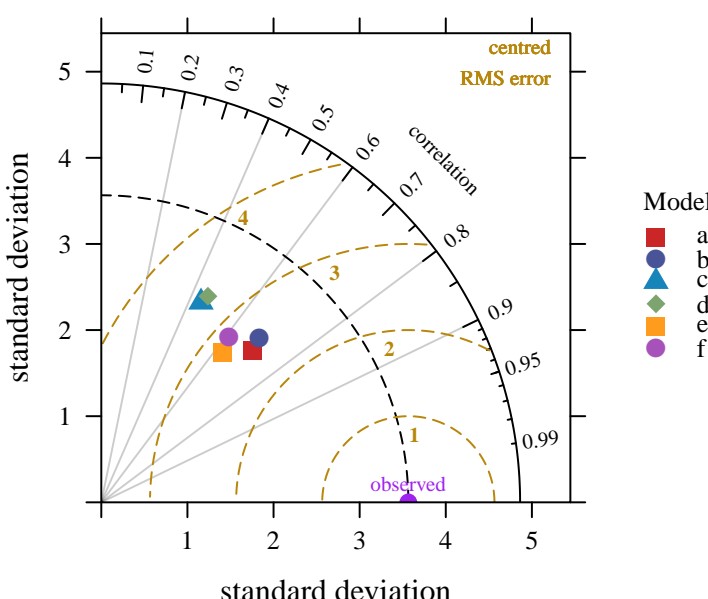

**Figure 6.** A Taylor diagram summarising the statistical comparisons between the modelled NH₃ fluxes from the six models a–f described in the text and the measured NH₃ fluxes.

Although the leaf wetness parameter measured with the leaf clip sensors is somewhat empirical, it is not surprising that it appears to reflect the actual leaf water amount more closely than the proxies via VPD and $RH$. All three parameters should be closely linked. Even after optimisation of the models, however, extensive differences in model output remain, principally between the models using leaf wetness parameter and $RH$ as model inputs. Figure 7 presents a scatter plot of leaf wetness measurements against $RH$ normalised to the canopy height. The relationship between them is best described through a power equation, which suggests that leaf wetness decreases more sharply than RH across the campaign. Indeed, Figure 2 shows a distinct lag between observed $RH$ (as well as VPD) and the leaf wetness parameter. While $RH$ minima are detected between 11:00 and 13:00, and ranges in a fairly narrow band between 100% and 80%, leaf wetness reaches minima between 13:00 to 16:00, and can decrease significantly, particularly during Period Two of the campaign.

Overall, the results indicate that there is significant value for interpreting field measurements in making direct measurements of leaf wetness using leaf wetness clip sensors of the type used here. Although leaf wetness or surface wetness measurements are not typically available for use in chemistry transport models, canopy wetness is often simulated by land surface models (Katata et al., 2010) and within chemical transport models (Campbell et al., 2019). Simulated surface wetness data, as a leaf

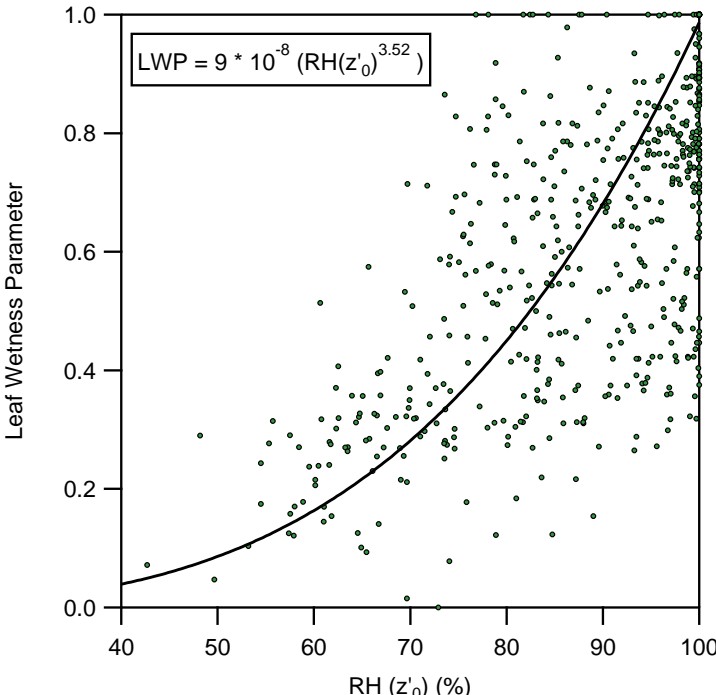

**Figure 7.** Scatter plot of hourly leaf wetness parameter and relative humidity $z_0'$ measurements taken during campaign, with a fitted power relationship.

wetness parameter, could therefore be used to model $NH_3$ fluxes using the LWP dependent $R_w$ parameter as described here. Alternatively, VPD-based parameterisations, over $RH$ based parameterisations, could be employed instead.

### 4.4 Model Performance with respect to the choice of stomatal emission potential

As is visually apparent in Figure 7, the influence of the apoplastic ratio is relatively minimal for reproducing flux variability in
comparison to the effect of $R_w$ parameterisation over the $\Gamma_s$ range explored (38.5 to 50). However, the choice of $\Gamma$ does affect the model's ability to reproduce the overall magnitude of the fluxes during daytime (Table 2).

Models that used the $\Gamma_s$ value of 50 (the upper bound to the inferred values of 38.5 for $\Gamma_s$) (b, d and f) simulated values better in agreement with observations in comparison to their paired $R_w$ models (respectively, a, c, and e) which used the value of 38.5. In particular, the use of 38.5 as a value led to models underestimating the scale of the emissions.

The discrepancy highlights a potential problem with using the method of inferring $\Gamma_s$ as outlined in Section 3.3 in tropical conditions. As outlined by Nemitz et al. (2004), the validity of equating $\chi_a$ to $\chi_c$ only holds for dry conditions (e.g. $RH <$ 50%), when $R_w$ can reliably be expected to be large. At higher humidity values, leaf cuticles may start to become a small sink and $\chi_c$ becomes an underestimate of $\chi_s$. At the ATTO site, where median humidity at the canopy level throughout the campaign was 87%, with only a few occurrences during the drier Periods Two and Four where it fell below 60%, this approach of inferring

$\Gamma_s$ from $NH_3$ measurements was likely affected. However, the impact does not appear to have completely invalidated the use of the method, as the somewhat larger value of 50 that resulted in models with best agreement still lies within one standard deviation of the inferred $\Gamma_s$ value. An accurate determination of apoplastic ratio for tropical rainforest, derived from leaf assays, would improve the accuracy of the model, and therefore remains an important area for future investigation. However, its variability across the large plant species diversity would likely provide a challenge in deriving a bottom-up mean value that

governs the net exchange.

### 4.5   Temporal variability in model performance

Modelled values diverge significantly from observations at several points during the campaign. In particular, on 30 October every model predicts an earlier, less sustained emission in comparison to the observation, while on 2 November, no model predicts any emission, contrary to observations, which suggest a strong emission of $NH_3$ from 13:00 to 15:00. With regards

to the divergence in models from the observations on 2 November, the possibility of other sources of $NH_3$ emission could be considered that would not be accounted for using the single layer model. For example, from the evening of 31 October to the early morning of 2 November, heavy precipitation periods were recorded, coupled with increased deposition fluxes of $NH_3$ on 1 November. Increased wet deposition of N through $NH_4^+$ in rainwater and washed from the canopy (Nemitz et al., 2000) to the forest floor or the higher soil moisture itself could have led to an increase in soil or leaf litter microbial activity below

canopy. Subsequent drying of the soil and leaf litter throughout 2 November might have led to an evaporation of $NH_3$ from the litter or soil layer from the forest floor (Hansen et al., 2017), leading to observed emissions of $NH_3$ in the afternoon. This potential scenario would not be modelled with the single layer canopy resistance model.

    Average modelled values for daytime tended to agree better with their corresponding period of observations than night-time values. Overall, the six models tended to overestimate nocturnal $NH_3$ deposition, particularly during Period Two and Four

when all six models overestimated the average deposition by more than 25% from the corresponding average observations. Of course, the flux measurement itself is not without error, especially during the calmer and more stable night-time conditions.

### 5   Conclusions

Observations of the bi-directional surface-atmosphere exchange of $NH_3$ at a tropical rainforest site have been successfully replicated using a static single-layer resistance model. Application of a capacitance model that additionally incorporates the

process of cuticular desorption did not lead to improved model results, suggesting that the emission periods were under stomatal control. Models that used a single layer canopy resistance approach, where the cuticular resistance was governed either by $RH$, VPD or a measurement of leaf wetness, were able to replicate the pattern of observed $NH_3$ deposition with frequent periods of afternoon emissions. Of all the models used, the most successful was a cuticular resistance modelling approach based on using leaf wetness measurements, and where modelled $\chi_c$ was governed by an apoplast $\Gamma = NH_4^+/H^+$ ratio of 50, one standard

deviation above the mean inferred from the measurements.

The periods when the most frequent emissions of $NH_3$ occurred, and which were most successfully modelled by cuticular resistance models, are typified by conditions that diverge from the overall campaign average, i.e., above-average temperatures, and below average relative humidity. This campaign took place in the dry season, and so comparison with the surface exchange of $NH_3$ during the wet season would be a necessary first step in determining if stomatal exchange is the principal driver of $NH_3$

surface exchange throughout the year. Long-term observations would also be required to determine whether the temperature increases, drought conditions and elevated ambient $NH_3$ concentrations that are anticipated from climate change and human development over this region have any impact on $NH_3$ surface exchange; and whether prolonged reactive nitrogen input from biomass burning activities raise compensation points.

One outcome of this study has been to establish the suitability of leaf wetness measurements, converted to a suitable pa-

520 rameter, as a factor for modelling cuticular resistance in $NH_3$ surface exchange modelling. Leaf wetness measurements, albeit from wetness grid sensors, have been used previously in $NH_3$ modelling, but these were first converted to an associated value of $RH$ before being used in $RH$-based $R_w$ parameterisations. Using leaf contact sensors, this study demonstrates that leaf wetness can be used directly, with a $R_w$ parameterisation that here proved to be the most sensitive and accurate in simulating cuticular $NH_3$ exchange.

A $\Gamma$ value of 50 led to the best modelling of $\chi_c$ values, and hence to the best fit with observed values. While within one standard deviation from the initially inferred value of 38.5, this did highlight that the method used to infer apoplastic ratio perhaps suffered under the high humidity conditions present at the rainforest site. An accurate determination of emission potential for this region is required for global scale modelling, necessitating accurate measurements of apoplast ratio i.e. the use of leaf assays to determine $\Gamma_s$. Future studies of $NH_3$ surface exchange above rainforest should therefore seek to incorporate

accurate determinations of leaf apoplastic ratio as a necessary part of their methodology.

Some periods of divergence between the models and observed values highlight that other sources of $NH_3$ surface exchange (such as soil or leaf litter exchange) should be incorporated into future investigation, while also emphasising the difficulty in measuring and modelling $NH_3$ surface exchange in remote, challenging conditions. A complete understanding of $NH_3$ surface exchange dynamics at rainforest sites will require a full suite of instruments measurements, incorporating in-canopy

measurements of $NH_3$ concentration gradients, trunk space flux measurements, and characterisation of leaf and soil $NH_4^+$ pools.

*Author contributions.* RR and CFDM took the measurements of $NH_3$. Data was processed by RR, CFDM, EN, MS and MH. Meteorological data used in the campaign was taken by MSá and AA. Leaf wetness measurements were taken byMS and RR. RR undertook the modelling of $NH_3$ with guidance from EN and aid from CFDM, MS and MH. RR wrote the manuscript, with subsequent contributions from all co-authors.

*Competing interests.* The authors declare that they have no conflict of interest.

*Acknowledgements.* RR acknowledges studentship funding jointly from the Max Planck Institute for Chemistry and the University of Edinburgh School of Chemistry. CDM, EN and the GRAEGOR instrument were supported through National Capability funding from the UK Natural Environmental Research Council (NERC), including through the UK-SCAPE project (NE/R016429/1) . We thank the Instituto Nacional de Pesquisas da Amazonia (INPA) and the Max Planck Society for continuous support. We acknowledge the support by the German Federal Ministry of Education and Research (BMBF contracts 01LB1001A and 01LK1602B) and the Brazilian Ministério da Ciência, Tecnologia e Inovação (MCTI/FINEP contract 01.11.01248.00) as well as the Amazon State University (UEA), FAPEAM, LBA/INPA and SDS/CEUC/RDS-Uatumã).

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
