# Peer review of "Measurement and modelling of the dynamics of NH3 surface-atmosphere exchange over the Amazonian rainforest"

_Biogeosciences, 2020_

## Referee Comment (RC1) · Anonymous Referee #1 · 22 Aug 2020

General Comments The authors demonstrated the results of high-resolution measurements of NH3 flux and wetness parameters in the Amazonian rainforest, where and then determined the NH3 emission potential and the parameterization of the best performance. The manuscript is well organized and has interesting results rarely investigated at a tropical forest. I generally agree with the conclusions, but one concern about the new parameterization of leaf wetness parameter (LWP) popped up in my mind. When the authors "focus(ed) on the most suitable control metric for Rw" (L70), they found that the parameterization as Eq. (17) has the statistically better result than those using RH and VPD. Nevertheless, the final message is kind of negative as "However, such a parameter is ... RH based parameterizations" (L431-432) and "However,

[Figure]

VPD may be the parameter of choice for chemistry transport models because it is more readily simulated." (L490-491). I can understand the meaning, but could you consider about much positive recommendation? For example, canopy wetness is actually often simulated by several land surface models coupled with meteorological model (e.g., Katata et al., 2010) and even in the chemical transport model (e.g., Cambell et al., 2019). In order to evaluate Eq. (17) for further model improvements, the comprehensive datasets of past and present (and future) LWP datasets in the world are helpful.

P.C. Campbell, J.O. Bash, and T.L. Spero (2019) Updates to the Noah Land Surface Model in WRF-CMAQ to Improve Simulated Meteorology, Air Quality, and Deposition. J. Adv. Model Earth. Syst., 11, 231–256. G. Katata, H. Nagai, M. Kajino, H. Ueda, and Y. Hozumi (2010) Numerical study of fog deposition on vegetation for atmosphere-land interactions in semi-arid and arid regions. Agr. Forest Meteorol., 150, 340-353.

Specific Comments L44, L138, L353, and references: no journal information of Ramsay et al. (2020) which includes the core data of this study. I believe that this must be peer-reviewed by appropriate scientific journal, but if it is not the case, please provide the detailed information about, e.g., the validity of the flux measurements of HNO3 and HCl (L138) and the estimation method of reactive N dry deposition, 1.74 kgN/ha/yr (L353).

Technical Corrections L24-27: the evidence from recent literature is required Fig. 2 and 3: it is hard for readers to compare the flux (Fig. 3) with related parameters of RH, VPD, and LWP (Fig.2). Why not separate Fig. 2 into two periods as Fig. 3 and combined them with Fig. 3? Fig. 4: pls specify the period, and add "(LWP)" just after "leaf wetness parameter ." Also x-axis of the third panel may be revised as "LWP" L301: insert a space between "alpha" and "for a fully ..." L309: nighttime should be "1900-0600" to cover the entire period of one day? L364-365: revised as "... openings (Urban et al., 2017), while ..." L385-387: the positions of several parentheses are wrong. also "(1998)" is duplicated L456: should be "heavy precipitation periods"

---

## Referee Comment (RC2) · Anonymous Referee #2 · 9 Oct 2020

Ramsay et al. analyze a month of dry-season observed NH3 fluxes in the Amazon with a popular NH3 surface exchange model. The observational dataset is very unique. They infer the ratio of apoplastic NH4+ to H+ concentrations from their observations, and examine how different parameterizations of leaf wetness influence the overall agreement between measured and modeled NH3 fluxes. The authors have a novel set of leaf wetness measurements that shows the importance of getting this variable correct in modeling NH3 surface exchange. Overall, the paper is very well written, and I think it will be an important contribution to the peer-reviewed literature provided the authors make some adjustments.

[Figure]

Major concerns Do the authors have an indication of how well the Wesely stomatal resistance model simulates stomatal resistance at ATTO? Is it consistent with an inversion of water vapor fluxes? I doubt Wesely captures much variability at all, which raises the question of how much can be inferred about cuticular resistance when the authors are likely not capturing much variability if at all with Wesely stomatal resistance? In general, I'd like to see more discussion of uncertainties in the flux, as well as in Ra, Rb, & Rs might affect the authors' results. I also didn't find the discussion of stomatal conductance increasing with leaf temperature to be convincing.

I would also like to see a more in-depth analysis than Figure 6 in arguing that there really is one 'best' model (i.e., I would like to see a figure that is more convincing that variability is better captured when leaf wetness observations are used).

Line comments Line 32 – how can bidirectional exchange have been considered a perfect sink? the subject of the modifier is incorrect Line 46 – is Ramsay the companion paper? please specify Line 67 – I'm not sure what 'this model' refers to – the authors have discussed several different types of model. Line 69-70 – please rephrase this sentence – the authors only present models that accurately simulate the observed fluxes? Line 79 – what is an aerodynamic canopy height? Line 85 – rephrase to something like 'NH3 fluxes can be considered representative of a homogenous rainforest' Line 86 – so the footprint of 5.2 km is for high wind? Line 119 – wind speed and direction were measured by eddy covariance? Line 130 – I think the authors are using two different terms to describe vertical concentration differences Line 247 – why don't the authors use the same form of Ra as used for the AGM? Line 150 – 'this canopy resistance approach'? what canopy resistance approach? do the authors mean their way of inferring Rc through residual of 1/vd and Ra + Rb? Line 163 – notional mean? can the authors provide a clearer definition of $\chi\_c$ Line 182 – conceptual mean? Line 190-3 – this is a very long sentence – will the authors split it up? Figure 2 – add shading for different periods Figure 3 – can this plot be four panels? it's hard to read as is. this plot is also not colorblind friendly Line 242 – I think the statistical summary is on Figure

4 Line 274 – 'furthermore' to what? can the authors spell out what they're implying here? Line 295 – is the parameterization novel? or is it novel that the authors actually have observations of leaf wetness? Line 335 – aren't the authors deriving fluxes from concentration measurements? perhaps clarify how the technique here is different from Trebs and Adon Line 344-7 – I guess I would have liked to see this information upfront (I was wondering throughout) Line 364-5 – Is this really true? Leaf temperatures usually accompany increases in vapor pressure deficit – but in Urban et al. 2017 VPD is controlled for. Line 363-370 – generally it was hard for me to follow what this paragraph is referring to – can the authors remind the reader what aspect of their results they are discussing? Line 373 – I think this statement is too strong Line 372 -9 – there is a lot of information here (from the sentence starting with 'Nevertheless' to the end of the paragraph), and I'm not sure much of it is needed except the last sentence. Line 388 – please remind the reader that emission potential = apoplastic ratio (or is it an inverse relation?) Line 392 – reference for values of ratio being as low as 5-10? Line 392-5 – this sentence is really long - please make the point clearer Line 395 – are the authors suggesting the soil is nitrogen-poor at ATTO? please clarify Line 395 – "impact" → "decrease" Line 397-9 – suggest removing explicit value judgement here Line 404 – suggest using the model simulation names here so the reader doesn't have to dig in the text for what a,b,c,d, etc are Line 405 – doesn't this make sense, given that the ratio is constant whereas Rw gives the estimate more variability (Ok, I see this is discussed later on, but the authors should consider including this info here, as well as reframing a little) Figure 6 – I don't really find this analysis all that convincing. can the authors really say one model is better than another? maybe if would be helpful if the authors highlighted better agreement when the leaves are wet. Line 415 – to some degree, doesn't the modeled value also have measurement uncertainty? because the authors are driving it w/ observations Line 423-4 – I'm confused by this sentence – aren't leaf wetness and RH model inputs? Line 432 – what about just a better RH-> leaf wetness parameterization? I think this sentence needs to be adjusted Line 440 – would be helpful to remind the reader here that the 38.5 is the inferred value, and the

50 is the upper bound of the inferred value Line 448-9 – I think this sentence makes the value sound more uncertain than it is Line 461-2 – it seems like the authors are saying the same thing twice in one sentence Line 467-9 – I'm not sure the authors can comment on whether these conditions diverged from mean climate – maybe just say campaign average – the observations are from 1 month of 1 year Line 471-2 – I would suggest cutting this sentence here – it makes your results seem questionable without more discussion of why the difference is ok (which the authors do well in the discussion) Line 476 – The phrase 'somewhat larger' is ambiguous – I would urge the authors to be more concrete with their wording here Line 479-1 – yes in that the authors could see how their best fit model from the dry season works for the wet season, but Wesely stomatal conductance is not going to capture differences between wet and dry Line 494-5 – can the authors be more concrete here? do they really not think

---

## Author Comment (AC1) · 5 Mar 2021

The comment was uploaded in the form of a supplement:
https://bg.copernicus.org/preprints/bg-2020-219/bg-2020-219-AC1-supplement.pdf

---

## Author Comment (AC2) · 5 Mar 2021

The comment was uploaded in the form of a supplement:
https://bg.copernicus.org/preprints/bg-2020-219/bg-2020-219-AC2-supplement.pdf

---

## Author Response (AR1)

**Authors' Response to Reviews of**

**Measurement and modelling of the dynamics of NH$_3$ surface-atmosphere exchange over the Amazonian rainforest**

R. Ramsay et al.

*Biogeosciences,* `doi.org/10.5194/bg-2020-219`
* * *
**RC:** *Reviewers' Comment*,    AR: Authors' Response,    ☐ Manuscript Text

**1. Introduction**

We thank the reviewers for their time in reviewing our paper and providing comments, and we apologise for the delay in our response. Below, we present our responses to each reviewer in turn, addressing each of their points individually. With some points, there is overlap between reviewers. Where this arises, we have included both comments, and provided a response.

All line numbers given in this response refer to those given in the original manuscript.

**2. Reviewer #1**

**2.1. General Comments from Reviewer #1**

**RC:** *I generally agree with the conclusions, but one concern about the new parameterization of leaf wetness parameter (LWP) popped up in my mind. When the authors "focus(ed) on the most suitable control metric for $R_w$" (L70), they found that the parameterization as Eq. (17) has the statistically better result than those using RH and VPD. Nevertheless, the final message is kind of negative as "However, such a parameter is...RH based parameterizations" (L431-432) and "However, VPD may be the parameter of choice for chemistry transport models because it is more readily simulated." (L490-491). I can understand the meaning, but could you consider about much positive recommendation? For example, canopy wetness is actually often simulated by several land surface models coupled with meteorological model (e.g.,Katata et al., 2010) and even in the chemical transport model (e.g., Cambell et al.,2019). In order to evaluate Eq. (17) for further model improvements, the comprehensive datasets of past and present (and future) LWP datasets in the world are helpful.*

**AR:** We thank the reviewer for their recommendation. We have therefore revised L431-432 to read: "Although leaf wetness or surface wetness measurements are not typically available for use in chemistry transport models, canopy wetness is often simulated by land surface models (e.g. Katata et al., 2010) and within chemical transport models (e.g. Campbell et al., 2019). Simulated surface wetness data, as a leaf wetness parameter, could therefore be used to model NH$_3$ fluxes using the LWP dependent $R_w$ parameter as described here." We have also removed line 491, which states that the VPD dependent $R_w$ parameterisation may be the preferred method for chemistry transport models.

**2.2. Specific comment**

**RC:** *L44, L138, L353, and references: no journal information of Ramsay et al. (2020) which includes the core data of this study. I believe that this must be peer-reviewed by appropriate scientific journal, but if it is not the case, please provide the detailed information about, e.g., the validity of the flux measurements of $HNO_3$ and HCl (L138) and the estimation method of reactive N dry deposition, 1.74 kgN/ha/yr(L353).*

**RC:** *(from Reviewer #2) Line 46 – is Ramsay the companion paper? please specify*

AR: We have now included an updated reference to Ramsay et al. (2020) (doi.org/10.5194/acp-2020-586) which presents the core data used within this paper.

**2.3. Technical Corrections**

**2.3.1 Revision 1**

**RC:** *L24-27: the evidence from recent literature is required*

AR: We have added a reference to the review by Fowler et al., (2013) (10.1098/rstb.2013.0164), which outlines the nitrogen cycle and the impacts of increased nitrogen deposition on nitrogen limited ecosystems.

**2.3.2 Revision 2**

**RC:** *Fig. 2 and 3: it is hard for readers to compare the flux (Fig. 3) with related parameters of RH, VPD, and LWP (Fig.2). Why not separate Fig. 2 into two periods as Fig. 3 and combined them with Fig. 3?*

**RC:** *(from Reviewer # 2) Figure 2 – add shading for different periods. Figure 3 – can this plot be four panels? it's hard to read as is. this plot is also not colorblind friendly*

AR: We have attempted to accommodate the suggestions of both reviewers. We have now combined Figure 2 and 3, into a new Figure 2. This figure shows the time series of measured and modelled $NH_3$ fluxes alongside time series of the meteorological parameters. Periods are now demarcated for all time series, with no shading. A colour blind friendly palette has been used for the measured and modelled $NH_3$ fluxes.

**2.3.3 Revision 3**

**RC:** *Fig. 4: pls specify the period, and add "(LWP)" just after"leaf wetness parameter ." Also x-axis of the third panel may be revised as "LWP"*

AR: We have corrected this.

**2.3.4 Revision 4**

**RC:** *L301: insert a space between "alpha" and "for a fully ..."*

AR: We have corrected this.

**2.3.5 Revision 5**

**RC:** *L309: nighttime should be "1900-0600" to cover the entire period of one day?*

**AR:** During the campaign, the authors took hourly measurements of $NH_3$ fluxes. There are therefore 24 flux measurements per day. For the purposes of dividing flux measurements into "day" and "night" periods, each consisting of 12 flux measurements and taking account of the local solar time at the ATTO site, we selected the period of 06:00 – 17:00 as day (12 flux measurements) and 18:00 – 05:00 as night (also 12 flux measurements). The start and end times for each period are inclusive of flux measurements.

**2.3.6 Revision 6**

**RC:** *L364-365: revised as "... openings (Urbanet al., 2017), while ..."*

**AR:** We thank the reviewer for noting this, and we have corrected for this.

**2.3.7 Revision 7**

**RC:** *L385-387: the positions of several parentheses are wrong. also"(1998)" is duplicated*

**AR:** We have corrected these errors.

**2.3.8 Revision 8**

**RC:** *L456: should be "heavy precipitation periods"*

**AR:** Corrected.

**3. Reviewer #2**

**3.1. Major concerns**

**RC:** *Do the authors have an indication of how well the Wesely stomatal resistance model simulates stomatal resistance at ATTO? Is it consistent with an inversion of water vapor fluxes? I doubt Wesely captures much variability at all, which raises the question of how much can be inferred about cuticular resistance when the authors are likely not capturing much variability if at all with Wesely stomatal resistance? In general, I'd like to see more discussion of uncertainties in the flux, as well as in $R_a$, $R_b$, & $R_s$ might affect the authors' results. I also didn't find the discussion of stomatal conductance increasing with leaf temperature to be convincing.*

**AR:** To address the issue of whether the Wesely function for calculating stomatal resistance is appropriate for our study, we have revised the section placement of the manuscript and added further material. Section 2.5 of the current manuscript is now Section 2.4, and includes further material on the derivation of $\varepsilon_{\mathrm{sat}}(T_{z_0'})$) and $\varepsilon_{z_0'}$. Section 2.5, originally Section 2.4, now addresses the concern over the appropriateness of the Wesely function for stomatal resistance. Considering the importance of this issue, we have reproduced verbatim the addition to the new Section 2.5. here:

The appropriateness of this parameterisation and the choice of parameter $R_i$ was evaluated against the water vapour fluxes that were measured during fairly dry conditions when stomatal evapotranspiration is expected to be the dominant source.

The bulk stomatal resistance ($R_{sb}$) for water exchange can be calculated from the measured water vapour flux ($F_{H_2O}$) as (Nemitz et al., 2009):

$$R_{sb} = \frac{\varepsilon_{\text{sat}}\left(T_{z_0'}\right) - \varepsilon_{z_0'}}{F_{H_2O}} \tag{1}$$

To avoid periods during which sources other than evapotranspiration contribute to the water flux we applied a stringent filter criterion to exclude periods during or within two hours of rainfall or with $RH>80\%$. This left 55 30-minute values for the assessment. Measurement-derived values of $R_{sb}$ for $H_2O$ were converted to the equivalent resistance for $NH_3$, accounting for the differences of the molecular diffusivities of the two gases (e.g. Hanstein et al., 1999), and the comparison was carried out on their reciprocal values (stomatal conductances, $G_s$ and $G_{sb}$), because it is the uncertainty in the stomatal conductances that propagates directly into the predicted flux. A linear regression analysis revealed a very high $R^2$ value of 0.97 and a slope of 0.95 (using an intercept of 0), suggesting that the modelled resistances were slightly larger, but well within the range of the measurement uncertainty of $R_{sb}$. Therefore, the parameterisation based on (Wesely, 1989) is appropriate for this site.

We have also included a subsection in the Results section that discusses the issue of errors. We have again reproduced the majority of the additional material here:

The error in the observed fluxes of $NH_3$ during this campaign are presented in Ramsay et al., 2020. Using a Gaussian error propagation approach, a median percentage error of 33% was calculated for the observed $NH_3$ fluxes.

The error in the simulated fluxes can also be determined using an error propagation method...the total, simulated flux ($F_t$) is the sum of the cuticular flux ($F_w$) and the stomatal flux ($F_s$). The total uncertainty in $F_t$, $\sigma_{F_t}$, can therefore be determined by:

$$\sigma_{F_t} = \sqrt{\sigma_{F_w}^2 + \sigma_{F_s}^2} \tag{2}$$

where $\sigma_{F_w}$ and $\sigma_{F_s}$ are the associated errors in, respectively, $F_w$ and $F_s$. In turn, $\sigma_{F_w}$ and $\sigma_{F_s}$ can be calculated using a Gaussian error propagation...which are dependent on the errors in $R_a$, $R_b$, $R_s$ and $R_w$ measurements.

While the error in $F_s$ will remain the same for all simulated values of $F_t$, the error in $F_w$ will vary with the choice of $R_w$ parameterisation. Thus, the error in $R_w$ is the primary variable that governs the differences in the error in $F_t$ between the simulated values. Using this framework, the error values were calculated for each $R_w$ and $\Gamma_s$ parametrisation for the simulated total flux. In the overall calculated total flux in Table 1, the associated error, in $ng\,m^{-2}\,s^{-1}$, is also shown.

We believe that quoting only the median errors for the simulated total flux, $F_t$, is sufficient, as this value is what is directly compared to both the observed flux, and other simulated $F_t$ values. The note on the importance of the choice on $R_w$ parameterisation leads into the main emphasis of the subsequent discussion section, while addressing the Reviewer's point on how errors in resistance parameterisations can affect the final simulated flux.

Revisions 21 to 25 address the issue of stomatal conductance variation with temperature.

**RC:** *I would also like to see a more in-depth analysis than Figure 6 in arguing that there really is one 'best' model (i.e., I would like to see a figure that is more convincing that variability is better captured when leaf wetness observations are used).*

AR: Figure 6 of the original manuscript by itself would not provide an answer as to which model performs best against observations. However, Figure 7 in the original manuscript is included as a further analysis of model performance, by considering other model performance statistics such as the root mean squared error and the standard deviation of each model. Figures 6 and 7 should therefore be considered in tandem, alongside Table 1, to determine the model that performs best against measurements. We have revised lines 418-419 to emphasise this.

**3.2. Line Revisions**

**3.2.1 Revision 1**

**RC:** *Line 32 – how can bidirectional exchange have been considered a perfect sink? the subject of the modifier is incorrect*

AR: We wished to emphasise the development of knowledge with regards to $NH_3$ fluxes over forests, by detailing how forests were once considered to be perfect sinks, to now being both sink and source. We have reworded line 32 to read as follows to make this point clearer: "Forests were once considered to be perfect sinks for ammonia (Duyzer et al., 1992), until bi-directional surface exchange of $NH_3$ -— i.e., deposition and emission —- was recorded in many studies of $NH_3$ fluxes from forests."

**3.2.2 Revision 2**

**RC:** *Line 46 – is Ramsay the companion paper? please specify*

**RC:** *(from Reviewer #1 ) L44, L138, L353, and references: no journal information of Ramsay et al. (2020) which includes the core data of this study. I believe that this must be peer-reviewed by appropriate scientific journal, but if it is not the case, please provide the detailed information about, e.g., the validity of the flux measurements of $HNO_3$ and HCl (L138) and the estimation method of reactive N dry deposition, 1.74 kgN/ha/yr(L353).*

AR: We have now included an updated reference to Ramsay et al. (2020) (doi.org/10.5194/acp-2020-586), which presents the core data used within this paper.

**3.2.3 Revision 3**

**RC:** *Line 67 – I'm not sure what 'this model' refers to – the authors have discussed several different types of model*

AR: We have reworded line 67 to clarify that "this existing model" refers to the static canopy compensation point model.

**3.2.4 Revision 4**

**RC:** *Line 69-70 – please rephrase this sentence – the authors only present models that accurately simulate the observed fluxes?*

**AR:** We have replaced the word "accurately" with "adequately" in line 70, and have added the following line to the end of the paragrap: ""We discuss other model frameworks, such as the dynamic CCP model, that could be used to simulate $NH_3$ bi-directional surface exchange in Section 4.1, with a focus on the simplicity and performance of the SCCP model as justification for not pursuing more complex models further."

**3.2.5 Revision 5**

**RC:** *Line 79 – what is an aerodynamic canopy height?*

**AR:** The use of the term "aerodynamic canopy height" was wrong in this context. The value of 37.5 m given is the mean canopy height at ATTO. We have deleted the word "aerodynamic" here.

**3.2.6 Revision 6**

**RC:** *Line 85 – rephrase to something like 'NH$_3$ fluxes can be considered representative of a homogenous rainforest'*

**AR:** We have reworded the sentence as suggested.

**3.2.7 Revision 7**

**RC:** *Line 86– so the footprint of 5.2 km is for high wind?*

**AR:** The fetch is the horizontal distance between surface areas of differing homogeneity, and has an impact on the chosen sampling height. As a guideline, the geometric mean sampling height above $d$ should be no more than the fetch distance divided by 100. Consequently, one can also work backwards and estimate the footprint covered by a known geometric mean sampling height above $d$. We therefore estimate, based on a mean geometric height above $d$ of 15.1 m, that our site requires a fetch distance of 1.5 km. As the area of surface homogeneity extends in a 5.5 km radius from the flux tower, our sampling height is appropriate.

**3.2.8 Revision 8**

**RC:** *Line 119 – wind speed and direction were measured by eddy covariance?*

**AR:** We thank the reviewer for noting this. Wind speed and wind direction were measured using sonic anemometry, while the friction velocity and sensible heat were measured using eddy covariance. We have deleted "via eddy covariance" from the sentence.

**3.2.9 Revision 9**

**RC:** *Line 130 – I think the authors are using two different terms to describe vertical concentration differences*

AR: We thank the reviewer for noting this. We have now kept consistent with the use of $\Delta_c$.

**3.2.10 Revision 10**

**RC:** *Line 247 – why don't the authors use the same form of $R_a$ as used for the AGM?*

AR: Line 147 and Equation 2 provides the version of $R_a$ that was used both in this study, and in the study by Ramsay et al., 2020 which uses the modified aerodynamic gradient method to calculate $NH_3$ fluxes. We have added a reference to Ramsay et al., 2020 to clarify this.

**3.2.11 Revision 11**

**RC:** *Line 150 – 'this canopy resistance approach'? what canopy resistance approach? do the authors mean their way of inferring $R_c$ through residual of $1/v_d$ and $R_a + R_b$?*

AR: By "this canopy resistance approach", we are referring to the method of inferring the canopy resistance outlined in Equation 2. We have cross-referenced this as follows – "this canopy resistance approach *as outlined in Eq. 2...*".

**3.2.12 Revision 12**

**RC:** *Line 163 – notional mean? can the authors provide a clearer definition of $\chi_c$*

AR: It is important to stress that the concept of the canopy compensation point concentration, $\chi_c$, is a notional one, being the concentration analogue of the canopy resistance, $R_c$. As described in Equation 10, it is a modelled value. For this reason, we have used the term "notional". We have revised the sentence to make it clear that the notional mean concentration at canopy height is the canopy concentration point.

**3.2.13 Revision 13**

**RC:** *Line 182 – conceptual mean?*

AR: Our response to Revision 12 also explains this question.

**3.2.14 Revision 14**

**RC:** *Line 190-3 –this is a very long sentence – will the authors split it up?*

AR: We have revised the sentence to the following – "The extended model calculated the $NH_3$ holding capacity by estimating the leaf water amount in relation to $RH$. The ammonia holding capacity was implemented into the resistance framework by considering it analogous to an electric capacitor. The charge of this "capacitor"

depended dynamically on previously deposited NH$_3$, and tended to be released as dew dried out in the morning."

**3.2.15 Revision 15**

**RC:** *Figure 2 – add shading for different periods. Figure 3 – can this plot be four panels? it's hard to read as is. this plot is also not colorblind friendly*

**RC:** *(from Reviewer #1) Fig. 2 and 3: it is hard for readers to compare the flux (Fig. 3) with related parameters of RH, VPD, and LWP (Fig.2). Why not separate Fig. 2 into two periods as Fig. 3 and combined them with Fig. 3?*

**AR:** We have attempted to accommodate the suggestions of both reviewers. We have now combined Figure 2 and 3, into a new Figure 2. This figure shows the time series of measured and modelled NH$_3$ fluxes alongside time series of the meteorological parameters. Periods are now demarcated for all time series, with no shading. A colour blind friendly palette has been used for the measured and modelled NH$_3$ fluxes.

**3.2.16 Revision 16**

**RC:** *Line 242 – I think the statistical summary is on Figure 4*

**AR:** We thank the reviewer for noting this error. We have now inserted a cross reference to Figure 4.

**3.2.17 Revision 17**

**RC:** *Line 274 – 'furthermore' to what? can the authors spell out what they're implying here?*

**AR:** Line 274 should read as an extension of the preceding sentences outlining the use of an enhanced value for the apoplastic ratio. We have therefore reworded Line 274 to read "By using an enhanced value for $\Gamma_s$..."

**3.2.18 Revision 18**

**RC:** *Line 295 – is the parameterization novel? or is it novel that the authors actually have observations of leaf wetness?*

**AR:** To the authors' knowledge, there are no other parameterisations of $R_w$ that are based on the use of a leaf wetness parameter. In addition, most measurements of wetness are conducted using wetness grids, which provide binary qualitative ("wet"/"dry") data that do not lend themselves to leaf wetness parameterisations, unlike the leaf clip method used in this study. We therefore maintain that this is a novel parameterisation for $R_w$.

**3.2.19 Revision 19**

**RC:** *Line 335 – aren't the authors deriving fluxes from concentration measurements? perhaps clarify how the technique here is different from Trebs and Adon*

AR: This is correct. The studies by Trebs et al., 2004 and Adon et al., 2013 infer fluxes from single point concentration measurements, using an inferential modelling approach, rather than using concentration gradients to derive the flux. We have revised line 335 to note this.

**3.2.20 Revision 20**

RC: *Line 344-7 – I guess I would have liked to see this information upfront (I was wondering throughout)*

AR: As the SCCP model has shown to perform adequately at the site, more complex models such as capacitance model are not be required to the simulate the flux. Furthermore, the simpler SCCP models offers the advantage over more complex models in that it can cope better with data gaps, and therefore cover a wider measurement range. As discussed in Revision 4, we have altered the wording to "adequately" rather than "accurately", and included the following: "We discuss other model frameworks, such as the dynamic CCP model, that could be used to simulate $NH_3$ bi-directional surface exchange in Section 4.1, with a focus on the simplicity and performance of the SCCP model as justification for not pursuing more complex models further."

**3.2.21 Revision 21**

RC: *Line 364-5 – Is this really true? Leaf temperatures usually accompany increases in vapor pressure deficit – but in Urban et al. 2017 VPD is controlled for.*

AR: We have revised line 364 to note that in the study by Urban et al., VPD was a control variable.

**3.2.22 Revision 22**

RC: *Line 363-370 – generally it was hard for me to follow what this paragraph is referring to – can the authors remind the reader what aspect of their results they are discussing?*

AR: We have revised line 363 to read: "The consistently warmer noon-time conditions at the leaf canopy during measurements at the ATTO site would also favour stomatal exchange driven emissions of $NH_3$."

**3.2.23 Revision 23**

RC: *Line 373 – I think this statement is too strong*

AR: See Revision #24

**3.2.24 Revision 24**

RC: *Line 372 -9 – there is a lot of information here (from the sentence starting with 'Nevertheless' to the end of the paragraph), and I'm not sure much of it is needed except the last sentence.*

AR: We agree and have now deleted material from the sentence beginning "Nevertheless..." to "...is not reached".

**3.2.25 Revision 25**

**RC:** *Line 388 – please remind the reader that emission potential = apoplastic ratio (or is it an inverse relation?)*

**AR:** The emission potential is the same as the apoplastic ratio. We have amended line 388 to highlight this, and have also added a parenthetical addition to line 168 in the revised manuscript that further clarifies this.

**3.2.26 Revision 26**

**RC:** *Line 392 – reference for values of ratio being as low as 5-10?*

**AR:** The reference to Hanstein et al., (1999) has been moved to the end of the line. In this study, Hanstein presents measurements of the apoplastic ratio of *Arrhenatherum elatius* in a semi-natural, nitrogen limited environment, arriving at a ratio of $7 \pm 40$.

**3.2.27 Revision 27**

**RC:** *Line 392-5 – this sentence is really long - please make the point clearer*

**AR:** We have made the following revision to the paragraph: "The species of vegetation is also critical (Mattsson et al., 2009). Plants which are reliant on mixed nitrogen sources (ammonium, nitrate, and organic N), and which are more reliant on root rather than shoot assimilation of nitrogen, exhibit lower apoplast ratios than nitrate reliant, shoot assimilating species (Hoffmann et al., 1992)."

**3.2.28 Revision 28**

**RC:** *Line 395 – are the authors suggesting the soil is nitrogen-poor at ATTO? please clarify*

**AR:** We have deleted lines 395-397 ("While the N source...concentrations"), and revised line 397 onward with the following: "The value of 38.5 which was inferred from measurements lies in the range of $\Gamma_s$ values exhibited by semi-natural vegetation with low N inputs, and in the lower range of overall forest values quoted by Massad et al. (2010)."

**3.2.29 Revision 29**

**RC:** *Line 395– "impact"→"decrease"*

**AR:** See Revision 28

**3.2.30 Revision 30**

**RC:** *Line 397-9 – suggest removing explicit value judgement here*

**AR:** See Revision 28.

**3.2.31 Revision 31**

**RC:** *Line 404 – suggest using the model simulation names here so the reader doesn't have to dig in the text for what a,b,c,d, etc are*

AR: We think that including the model names here would overburden the text. However, as a reminder to the reader, we have inserted a cross reference to Table 1.

**3.2.32 Revision 32**

**RC:** *Line 405 – doesn't this make sense, given that the ratio is constant whereas $R_w$ gives the estimate more variability (Ok, I see this is discussed later on, but the authors should consider including this info here, as well as reframing a little)*

AR: We believe that the current placement of the discussion regarding how model variability is impacted by $R_w$ and $\Gamma_s$ is coherent within the flow of the text, and that introducing these points earlier in the text would detract from outlining the tests of model performance presented in Figures 6 and 7.

**3.2.33 Revision 33**

**RC:** *Figure 6 – I don't really find this analysis all that convincing. can the authors really say one model is better than another? maybe if would be helpful if the authors highlighted better agreement when the leaves are wet.*

AR: We think that this issue has been resolved in our response to the second of the major concerns raised by the reviewer.

**3.2.34 Revision 34**

**RC:** *Line 415 – to some degree, doesn't the modeled value also have measurement uncertainty? because the authors are driving it w/ observations*

AR: That is correct, and is addressed in our response to the first major concern. As mentioned in that response, the models are based on observations, and therefore they also have a degree of associated measurement uncertainty. We have revised the text to reflect this: "It should be borne in mind, however, that the standard deviation of the measured flux includes measurement uncertainty in addition to real variability, as do the modelled values, which are based on measured parameters."

**3.2.35 Revision 35**

**RC:** *Line 423-4 – I'm confused by this sentence – aren't leaf wetness and RH model inputs?*

AR: Leaf wetness and RH are model inputs for the determination of $R_w$. We have revised the sentence to read: "Even after optimisation of the models, however, extensive differences in model output remain, principally between the models using leaf wetness parameter and $RH$ as model inputs."

**3.2.36 Revision 36**

**RC:** *Line 432 – what about just a better RH->leaf wetness parameterization? I think this sentence needs to be adjusted*

AR: Our $RH$ based $R_w$ parameterisation is the result of least squares regression using the observed dataset. Within the framework of the the SCCP model, it cannot be further improved. Certainly, using a different model could improve the ability of $RH$, as a variable, to predict $NH_3$ fluxes, but we focus only on SCCP.

**3.2.37 Revision 37**

**RC:** *Line 440 –would be helpful to remind the reader here that the 38.5 is the inferred value, and the 50 is the upper bound of the inferred value*

AR: We have added a parenthetical comment that reminds the reader of this fact.

**3.2.38 Revision 38**

**RC:** *Line 448-9 – I think this sentence makes the value sound more uncertain than it is*

AR: We have revised the sentence to now read: "An accurate determination of apoplastic ratio for tropical rainforest, derived from leaf assays, would improve the accuracy of the model, and therefore remains an important area for future investigation. However, its variability across the large plant species diversity would likely provide a challenge in deriving a bottom-up mean value that governs the net exchange."

**3.2.39 Revision 39**

**RC:** *Line 461-2 – it seems like the authors aresaying the same thing twice in one sentence*

AR: We have revised the sentence to read "This potential scenario would not be modelled with the single layer canopy resistance model."

**3.2.40 Revision 40**

**RC:** *Line 467-9 – I'm not sure the authors can comment on whether these conditions diverged from mean climate – maybe just say campaign average – the observations are from 1 month of 1 year*

AR: We have revised the sentence to read "campaign average" instead of "expected climate".

**3.2.41 Revision 41**

**RC:** *Line 471-2 – I would suggest cutting this sentence here – it makes your results seem questionable without more discussion of why the difference is ok (which the authors do well in the discussion)*

AR: We thank the reviewer for this suggestion. We have deleted this sentence.

**3.2.42 Revision 42**

**RC:** *Line 476 – The phrase 'somewhat larger' is ambiguous – I would urge the authors to be more concrete with their wording here*

AR: We thank the reviewer for noting this. We have revised to read "one standard deviation above the mean inferred from the measurements".

**3.2.43 Revision 43**

**RC:** *Line 479-1 – yes in that the authors could see how their best fit model from the dry season works for the wet season, but Wesely stomatal conductance is not going to capture differences between wet and dry*

AR: We believe that this issue is addressed in our response to the first major concern.

**3.2.44 Revision 44**

**RC:** *Line 494-5 – can the authors be more concrete here?*

AR: We have added the following to the sentence: "...i.e. the use of leaf assays to determine $\Gamma_s$"